# Relative Preference Optimization: Enhancing LLM Alignment through Contrasting Responses across Identical and Diverse Prompts

## Abstract

In the field of large language models (LLMs), aligning models with the diverse preferences of users is a critical challenge. Direct Preference Optimization (DPO) has played a key role in this area. It works by using pairs of preferences derived from the same prompts, and it functions without needing an additional reward model. However, DPO does not fully reflect the complex nature of human learning, which often involves understanding contrasting responses to not only identical but also similar questions. To overcome this shortfall, we propose Relative Preference Optimization (RPO). RPO is designed to discern between more and less preferred responses derived from both identical and related prompts. It introduces a contrastive weighting mechanism, enabling the tuning of LLMs using a broader range of preference data, including both paired and unpaired sets. This approach expands the learning capabilities of the model, allowing it to leverage insights from a more varied set of prompts. Experiments in both paired and unpaired dataset settings, including tasks like dialogue, summarization, and general evaluation benchmarks, demonstrate RPO's superior ability to align LLMs with user preferences and enhance adaptability during training.

## 1 Introduction

Large language models (LLMs) such as ChatGPT OpenAI (2023) and LLaMA Touvron et al. (2023) have revolutionized AI, demonstrating remarkable capabilities in natural language processing, logical reasoning, and programming Pan et al. (2023); Tian et al. (2023). Their proficiency in zero-shot and few-shot learning is attributed to training on extensive, unsupervised datasets. However, the diverse nature of these datasets can result in alignment challenges, leading to outputs that may not consistently align with specific human values, particularly in nuanced contexts Agrawal et al. (2023); Shi et al. (2023); Liang et al. (2021); Sheng et al. (2019); Kadavath et al. (2022); Srivastava et al. (2022); Thoppilan et al. (2022); Bubeck et al. (2023). The Direct Preference Optimization (DPO) method fine-tunes the language model's policy to align more closely with human preferences, thereby eliminating the need for a separate reward model, a staple in traditional Reinforcement Learning from Human Feedback (RLHF) Schulman et al. (2017). Central to DPO is the utilization of pairwise preferences, with preferred and dispreferred responses identified for each prompt. This forms the foundation for effectively optimizing preferences. However, training a model to learn from an individual preference pair for each example may not fully capture the complexity of human learning. Human cognition often involves interpreting divergent responses, not only to identical questions but also to similar ones, highlighting the multifaceted nature of comprehension and preference formation Dahlin et al. (2018). Moreover, obtaining pairwise preference data can pose challenges and incur substantial costs, especially in sensitive domains such as healthcare and personal services, where careful attention to ethical considerations is essential Murtaza et al. (2023).

Our inspiration draws from the human learning process, where valuable insights often arise from the comparison of successful examples and relevant failures Dahlin et al. (2018). To emulate this, we introduce Relative Preference Optimization (RPO). This approach involves analyzing prompt similarities at the semantic level within each mini-batch, allowing us to classify pairs as either highly related or unrelated. We construct a contrast matrix that instructs the model to distinguish between preferred and dispreferred responses, applicable to both identical and semantically related prompts.

Figure 1: An example illustrates how DPO and RPO utilize contrastive responses with human preferences to achieve model alignment.

We have developed three weighting strategies to recalibrate the comparison of each contrastive pair. Our findings reveal that reweighting based on prompt similarities significantly enriches model alignment with human preferences, offering a more nuanced understanding. Furthermore, RPO inherently excels in handling non-pairwise preference data by considering semantically related contrastive pairs.

As illustrated in Figure 1, we are interested in the question "Explain the concept of photosynthesis." DPO applies penalties for incorrect responses and rewards for precise responses generated for the same prompt. Conversely, our method RPO emphasizes the semantic connections between various prompts. For instance, the prompt "Describe the importance of sunlight in plant growth" is conceptually similar, and its responses might intersect with those of the initial question. Under RPO, if an answer is less preferred for the second prompt, it is also treated as less suitable for the first prompt. Thus, RPO penalizes both $y_{l,1}$ and $y_{l,2}$ while approving $y_{w,1}$. It is crucial to note that not all prompts are semantically related enough to form effective contrastive pairs. RPO incorporates a reweighting mechanism, whereby unrelated prompts are given less emphasis during training. RPO expands the learning horizon of the model, empowering it to leverage insights from a broader range of prompts, mirroring the human learning process more closely.

We empirically evaluate RPO on several LLMs, including the LLaMA series and Mistral-7B, comparing it with SoTA preference alignment methods. RPO significantly outperforms the baselines on dialogue and summarization tasks, as well as evaluations on the general chat evaluation benchmarks.

The core contributions of RPO are summarized as follows:

- Innovative contrastive preference learning strategy: RPO enriches the landscape of preference optimization with novel contrastive learning techniques.

- Adaptability across varied contexts: Exhibiting exceptional adaptability, RPO is adept across a multitude of scenarios, whether or not explicit preference pairs are present, confirming its utility as a versatile tool in language model applications.

- Enhanced performance in critical language tasks: Demonstrating superiority over established methods like DPO, IPO, and KTO, the proposed RPO excels in key language processing tasks, including text summarization and dialogue generation, showcasing its improved alignment with human preferences.

## 2 RELATED WORK

The field of large language models (LLMs) has made notable strides in aligning these models with human preferences Chung et al. (2022); Schulman et al. (2017); Zhao et al. (2023); Rafailov et al.

(2023); Azar et al. (2024); Ethayarajh et al. (2024); Cheng et al. (2023); Pal et al. (2024), driven by innovative fine-tuning methodologies. In this exploration, we discuss several works closely related to our research.

## 2.1 REINFORCEMENT LEARNING FINE-TUNING (RLHF)

RLHF builds upon the foundation of SFT, employing RL to better align the model with human preferences Ouyang et al. (2022). The initial phase of RLHF involves learning a reward model from human preference data. This process typically utilizes the Bradley-Terry model Bradley & Terry (1952), which assesses the reward $r^*(y|x)$ for generating a specific response $y$ to a prompt $x$. The Bradley-Terry model determines the preference probability as follows:

$$p(y_w \succ y_l|x) = \frac{\exp(r^*(y_w|x))}{\exp(r^*(y_w|x)) + \exp(r^*(y_l|x))} = \sigma(r^*(y_w|x) - r^*(y_l|x)), \quad (1)$$

where $\sigma(\cdot)$ is the sigmoid function, and $r^*(y_w|x)$ and $r^*(y_l|x)$ represent the estimated rewards for the preferred and less preferred response, respectively, given prompt $x$. The loss function for the reward model, parameterized as $r_\phi$, is derived from a dataset $D$ of preference pairs:

$$L_R(\phi, D) = -\mathbb{E}_{(x,y_w,y_l)\sim D}[\log \sigma(r_\phi(x, y_w) - r_\phi(x, y_l))]. \quad (2)$$

The next phase is the RL fine-tuning process that seeks to optimize the policy $\pi_\theta$ based on the trained reward model. The objective is to maximize the expected reward, keeping the policy $\pi_\theta$ closely aligned with a reference model $\pi_{\text{ref}}$, usually derived from the SFT model. This optimization integrates KL-regularization to mitigate overfitting and preserve response diversity:

$$\max_{\pi_\theta} \mathbb{E}_{x\sim D, y\sim \pi_\theta(y|x)}[r_\phi(x, y)] - \beta \text{KL}[\pi_\theta(y|x)||\pi_{\text{ref}}(y|x)],$$

where $\beta$ is a scaling parameter. The PPO algorithm Schulman et al. (2017) is employed to iteratively update $\pi_\theta$ by estimating gradients and collecting new data from the current policy and reward model. A notable challenge in RLHF is managing the discrete nature of language generation, which complicates gradient back-propagation from the reward function to the policy.

## 2.2 DIRECT PREFERENCE OPTIMIZATION (DPO)

DPO Rafailov et al. (2023) offers an efficient approach by directly aligning a language model with human preferences, thus eliminating the need for a separate reward model. Utilizing direct human feedback, DPO refines the policy $\pi_\theta$ to better match nuanced human preferences. The objective of DPO is formulated as the following pairwise loss:

$$L_{DPO}(\pi_\theta; \pi_{ref}) = -\mathbb{E}_{(x,y_w,y_l)\sim D}\left[\log \sigma(\beta \log \frac{\pi_\theta(y_w|x)}{\pi_{ref}(y_w|x)} - \beta \log \frac{\pi_\theta(y_l|x)}{\pi_{ref}(y_l|x)})\right], \quad (3)$$

where $\beta$ is a scaling factor. DPO derives its reward function from the relationship between the policy $\pi_\theta$ and the reference model $\pi_{ref}$, with the inclusion of a partition function $Z(x)$ that normalizes the reward:

$$r(x, y) = \beta \log \frac{\pi_\theta(y|x)}{\pi_{ref}(y|x)} + \beta \log Z(x). \quad (4)$$

DPO's primary benefit lies in its stable training process, providing a more direct means of aligning models with human preferences. However, DPO's applicability is somewhat limited as it strictly defines its loss function based on the reward difference between chosen and rejected responses originating from the same prompt.

## 2.3 IDENTITY PREFERENCE OPTIMIZATION (IPO)

Identity Preference Optimization (IPO) Azar et al. (2024) addresses the overfitting challenge within the DPO framework. IPO introduces a regularization term into the DPO's loss function to maintain a balance between optimizing for human preferences and generalizing beyond the training data. The IPO loss function is expressed as:

$$L_{IPO}(\theta, y_w, y_l) = \left(\left(\log \frac{\pi_\theta(y_w|x)}{\pi_{\text{ref}}(y_w|x)} - \log \frac{\pi_\theta(y_l|x)}{\pi_{\text{ref}}(y_l|x)}\right) - \frac{1}{2\beta}\right)^2, \quad (5)$$

where $\beta$ serves as a regularization parameter. IPO enhances the training process by ensuring a more balanced response selection, contributing to the robustness of preference-based language models.

## 2.4 KAHNEMAN-TVERSKY OPTIMIZATION (KTO)

Kahneman-Tversky Optimization (KTO) Ethayarajh et al. (2024) diverges from the preference likelihood maximization used in DPO. Instead, KTO focuses on maximizing the utility of model outputs, informed by the human value function derived from Kahneman-Tversky's prospect theory. This adaptation to language models allows KTO to operate without the necessity of preference pairs, thereby streamlining data requirements. The KTO loss function is formalized as:

$$L_{KTO}(\pi_\theta, \pi_{ref}) = \mathbb{E}_{x,y \sim D}[w(y)(1 - h(x, y; \beta))], \tag{6}$$

where

$$h(x, y; \beta) = \begin{cases} \sigma(g(x, y; \beta)) & \text{if } y \text{ is desirable given } x \\ \sigma(-g(x, y; \beta)) & \text{if } y \text{ is undesirable given } x \end{cases} \tag{7}$$

$$g(x, y; \beta) = \beta \log \frac{\pi_\theta(y|x)}{\pi_{\text{ref}}(y|x)} - \mathbb{E}_{x' \sim D}\left[\beta \text{KL}(\pi_\theta \| \pi_{\text{ref}})\right]. \tag{8}$$

$w(y)$ represents the weighting factor applied to the KTO loss function. By default, $w(y) = 1$. KTO operates in unpaired scenarios by independently processing chosen and rejected samples.

## 3 RELATIVE PREFERENCE OPTIMIZATION

The traditional DPO framework aligns language model outputs with human preferences using pairwise data, where each pair is composed of a preferred (win) and dispreferred (lose) sample for the same prompt. However, this approach is limited to situations where such pairwise preference data is accessible, failing to exploit the valuable comparative insights that could be derived from contrasting diverse samples across a range of prompts. In response, our RPO framework encompasses a wider array of preference data, including non-paired samples. This development not only improves the use of existing preference data but also facilitates model training in complex scenarios where pair-wise data is not readily obtainable. More specifically, RPO integrates preference pairs derived from prompts that are semantically related but not identical, as shown in Figure 1. Through dynamic calculation of relative reward weights based on prompt similarities, our method enhances the model's ability to learn from a wider array of human feedback, resulting in better preference alignment.

### 3.1 CONTRAST MATRIX CONSTRUCTION

In RPO, the contrast matrix is a pivotal component that facilitates the comparison between win and lose responses to derive meaningful insights for model training. As shown in Figure 2, the construction of this matrix varies depending on whether the available data is paired or unpaired, allowing for flexibility in training dynamics.

**Paired Data Scenario.** In situations where each win response is associated with a corresponding lose response from the same prompt, the contrast matrix is an $M \times M$ square matrix, where $M$ represents the total number of unique prompts from a specific mini-batch within the dataset. Each element $c_{ij}$ within this matrix represents the contrastive score between the win response of the $i^{th}$ prompt and the lose response of the $j^{th}$ prompt. For diagonal elements where $i = j$, the score reflects the direct comparison within the same prompt, while off-diagonal elements represent the relative reward differences across distinct prompts. In this context, DPO is limited to using only the diagonal terms of the contrast matrix, while RPO takes into account all pairings within the matrix, encompassing a broader range of preference comparisons.

**Unpaired Data Scenario.** In cases where the dataset contains unpaired win and lose responses, the contrast matrix transforms into an $M \times N$ rectangular structure. Within this matrix, $M$ and $N$ respectively represent the number of unique win and lose samples in a batch of the dataset. Each element $c_{ij}$ in this matrix now indicates the contrastive score between the $i^{th}$ win response and the $j^{th}$ lose response, without the constraint of originating from the same prompt. This allows for a more extensive range of comparisons, as any win samples can be contrasted with any lose samples, harnessing the thematic connections within the dataset to enrich the model's preference learning.

For each win response $y_{w,i}$ and lose response $y_{l,j}$, the contrastive score $s_{ij}$ is computed as the difference in rewards associated with each response, defined by the following equations:

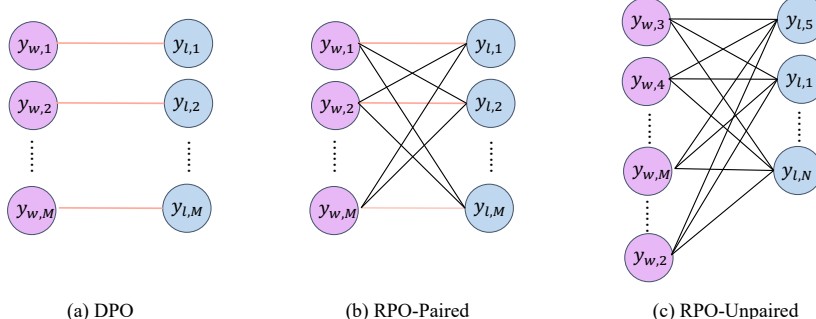

(a) DPO                (b) RPO-Paired                (c) RPO-Unpaired

Figure 2: DPO requires paired preference data derived from identical prompts. RPO can utilize preference data from either the same or different prompts for constructing contrastive samples. Here, $y_w$ represents win responses, and $y_l$ denotes lose responses.

$$s_{ij} = r_{w,i} - r_{l,j}, \quad \begin{cases} r_{w,i} = \beta \log \frac{\pi_\theta(y_{w,i}|x_i)}{\pi_{ref}(y_{w,i}|x_i)} + \beta \log Z(x_i), \\ r_{l,j} = \beta \log \frac{\pi_\theta(y_{l,j}|x_j)}{\pi_{ref}(y_{l,j}|x_j)} + \beta \log Z(x_j), \end{cases} \quad (9)$$

where $r_{w,i}$ denotes the reward associated with the $i^{th}$ win response, and $r_{l,j}$ signifies the reward for the $j^{th}$ lose response. By calculating the contrastive scores across the matrix, RPO enables a comprehensive evaluation of the relative preference among all potential contrastive pairs.

Similar to DPO, we define $Z(x) = \sum_y [\pi_{ref}(y|x) \exp(\frac{1}{\beta} \cdot r(x,y))]$. The normalization term $Z(x)$ in DPO is considered a constant that could generally be omitted Rafailov et al. (2023), so we assume that $Z(x)$ is constant for all prompts. We leave a detailed discussion regarding the normalization term for DPO-based method in Appendix A. Note the relative contrast idea could be easily extended to other preference learning algorithms, such as SimPO(Meng et al., 2024), where the reward model is defined without incorporating $Z(x)$. In Appendix C, we also extend our idea to SimPO. Next, we introduce how to weight each relative contrast pair.

### 3.2 Weighting Strategies

After forming the contrast matrix for each mini-batch in RPO, we deploy diverse strategies to assign differentiated weights to each comparison pair. These weights crucially determine the relative influence of different comparison pairs in the final loss computation. We propose a strategy that reweights the contrast matrix based on the distance between prompt feature embeddings for preference learning. For distinct data configurations, we have introduced two additional, simpler weighting strategies alongside our primary Embedding Distance Reweighting Strategy.

**Embedding Distance Reweighting Strategy.** The incorporation of prompt similarity plays a pivotal role in contrastive analysis, applicable to both paired and unpaired datasets. This technique involves calculating the cosine distance $d = \cos(f(x_w), f(x_l))$ between the embeddings of win ($x_w$) and lose ($x_l$) prompts, effectively assessing their thematic relatedness. This similarity directly influences the weight $\tilde{\omega}$ assigned to each pair of responses:

$$\tilde{\omega} = \exp\left(-\frac{d}{\tau}\right) = \exp\left(-\frac{\cos(f(x_w), f(x_l))}{\tau}\right). \quad (10)$$

Here, $f$ represents the model for extracting sentence embeddings, such as all-MiniLM-L6-v2 Wang et al. (2020), and $\tau$ acts as a temperature parameter that moderates the impact of prompt similarity on the weight. Specifically, a lower $\tau$ results in a greater variation in weights for different levels of similarity, emphasizing the contrastive scores between closely related prompts. In contrast, a higher $\tau$ leads to a more uniform distribution of weights, diminishing the disparity in influence between prompts of varying thematic similarity. The adjusted contrastive score $s_{ij}$ for a win and lose response pair is then calculated as:

$$s_{ij} = \omega_{ij} \times (r_{w,i} - r_{l,j}), \quad \omega_{ij} = \frac{\tilde{\omega}_{ij}}{\sum_{j'=1}^{N} \tilde{\omega}_{ij'}} \quad (11)$$

where $N$ represents the number of lose responses in the mini-batch, and each $\omega_{ij}$ is normalized within its respective mini-batch so that $\sum_{j=1}^{N} \omega_{ij} = 1$. In this configuration, the sum of the weights in each row of the matrix equals one.

We then obtain

$$s_{ij} = \omega_{ij} \times \beta \left( \log \frac{\pi_\theta(y_{w,i}|x_i)}{\pi_{ref}(y_{w,i}|x_i)} - \log \frac{\pi_\theta(y_{l,j}|x_j)}{\pi_{ref}(y_{l,j}|x_j)} \right). \tag{12}$$

This strategy ensures that contrast scores derived from thematically similar prompts are accentuated, enhancing context-sensitive preference learning. While the Embedding Distance Reweighting Strategy serves as the primary method to construct RPO's contrast matrix, we have explored two additional strategies for specific data configurations, offering alternative ways to weight the contrast matrix.

**Uniform Weighting Strategy.** Uniform Weighting can occur when the similarity between all prompt pairs is considered uniform or when the temperature parameter $\tau$ is set to positive infinity. Under this strategy, for an $M \times N$ contrast matrix, each winning response, compared against $N$ losing responses, is uniformly weighted at $1/N$ per comparison. This method simplifies the analysis and can be applied in both unpaired and paired data settings.

**Diagonal Emphasis Weighting Strategy.** This strategy can only be applied to paired data scenarios with an $M \times M$ contrast matrix. Central to this approach is the weighting factor $\alpha$, which crucially balances the impact of diagonal and off-diagonal elements in the matrix. Diagonal terms (where $i = j$) represent direct comparisons of win and lose responses for the same prompt, while non-diagonal terms account for comparisons across different prompts:

$$s_{ij} = \begin{cases} \alpha \times (r_{w,i} - r_{l,j}) & \text{if } i = j \\ \frac{(1-\alpha)}{M-1} \times (r_{w,i} - r_{l,j}) & \text{if } i \neq j \end{cases} \tag{13}$$

where $r_{w,i} - r_{l,j}$ follows the same formulation as described in Eq. 12.

The final RPO loss can be expressed as:

$$L_{\text{RPO}} = -\frac{1}{M \times N} \sum_{i=1}^{M} \sum_{j=1}^{N} \log \sigma \left( s_{ij} \right) \tag{14}$$

where $s_{ij}$ represents the adjusted contrastive scores calculated using one of the three weighting strategies mentioned before. This loss function directs the model to amplify the reward for winning responses and diminish it for losing ones among both identical and semantically related prompts. The learning intensity is dynamically modulated by both the prompt-aware reweighting factor $w$ and the scaling factor $\beta$.

## 4 EXPERIMENTS

We have undertaken a comprehensive series of experiments to address three primary questions of RPO: (a) Can conceptually related prompts be deemed effective contrastive pairs for human preference optimization? (b) What factors influence the performance of RPO? (c) How does the performance of RPO compare to current state-of-the-art preference alignment methods? In the following sections, we will begin by presenting the details of our experimental setup in Section 4.1. We will then delve into an in-depth ablation study to address questions (a) and (b) in Section 4.2, and finally, in Section 4.3, we will showcase the benchmark performance of our approach.

### 4.1 EXPERIMENTAL SETUP

**Training Tasks and Datasets.** Our experiments were conducted on three datasets to assess specific capabilities in summarization and open-ended text generation tasks.

Anthropic's Helpful and Harmless (HH) Dataset Bai et al. (2022): This paired dataset was utilized for assessing single-turn dialogue performance of our models. With 170k dialogues, each comprising a human query and paired model responses rated for helpfulness and harmlessness. Following DPO

Rafailov et al. (2023), the preferred responses from this dataset were utilized for the supervised Fine-Tuning (SFT) phase, aligning the initial model behavior with desirable conversational outcomes.

OpenAI's Summarization Dataset Stiennon et al. (2020), targeted for the summarization task, consists of 92,858 paired training samples. Each input $x$ in the dataset is a substantive forum post, and the task for the model is to generate a concise summary $y$. Similar to the HH dataset, the SFT phase was informed by preferred responses from this dataset, which set a benchmark for the model's summarization capabilities.

Binarized Datasets: The Binarized Capybara Dataset[1] contains 15.1k training samples derived from the Capybara-DPO 7K binarized dataset and consists of unpaired preference entries. Each entry in this dataset is a triplet (prompt, response, and a label indicating whether the response is deemed good or bad). The Binarized Ultrafeedback Dataset [2] includes 231k training samples for single-turn dialogue tasks. Currently, aside from KTO, other baseline algorithms do not accommodate unpaired preference datasets. Therefore, we conducted comparisons exclusively with KTO on these datasets.

**Baselines.** We assessed RPO against a range of alignment methods. These included SFT Chung et al. (2022) for initial model adaptation, PPO Schulman et al. (2017) for reinforcement learning fine-tuning, DPO and IPO Azar et al. (2024) for preference-based model alignment, and KTO Ethayarajh et al. (2024) as an alternative approach incorporating human value functions. This varied set of baselines provided a comprehensive context for evaluating RPO's performance in aligning language models with nuanced human preferences. For these comparisons, we utilized a range of pre-trained large language models, including LLaMA series Touvron et al. (2023) and Mistral-7B Jiang et al. (2023). For paired preference datasets, We conducted the evaluations of RPO on the validation sets of Anthropic's HH Dataset for dialogue and the OpenAI Summarization Dataset for summarization. To further challenge RPO's adaptability and conversation capability, we integrated the AlpacaEval leaderboard Li et al. (2023) into our evaluation benchmark. This benchmark comprises a set of 805 diverse and carefully curated prompts, serving as an ideal platform for testing the model's ability to follow general user instructions accurately and effectively. For the unpaired dataset, we also evaluated the performance of alignment algorithms trained on these two datasets across multiple benchmarks: the Open LLM Leaderboard Beeching et al. (2023) for question-answering, the MT-Bench Zheng et al. (2024) for assessing multi-turn conversational abilities, and the more challenging Arena-Hard benchmark Li et al. (2024), which evaluates chat capabilities using complex, real-world user queries.

Our primary evaluation metric was the win rate, calculated using the advanced capabilities of GPT-4 OpenAI (2023) as the evaluative tool. This metric quantitatively assessed the preference rate of our model's responses against those generated by baseline models. By employing GPT-4 for evaluation, we leveraged its robust understanding and judgment abilities as a stand-in for human evaluators Zheng et al. (2023); Li et al. (2023).

**Training Details.** For the paired dataset setting, preference data is processed into mini-batches, each comprising $N$ triplets $(x, y_w, y_l)$. It is important to emphasize that the volume of training data utilized in RPO are identical to those in DPO and KTO. For the unpaired dataset, each mini-batch consists of $(x, y, 0/1)$, representing prompts, responses, and their binary classification. This enables the construction of an $N_1 \times N_2$ contrast matrix for RPO fine-tuning, where $N_1$ and $N_2$ represent the number of chosen and rejected responses within the mini-batch, respectively.

In all experiments, both RPO and the baseline consistently utilized a beta value ($\beta = 0.1$) and a sampling temperature of 0. The training utilized 8 Nvidia A100 GPUs, with a batch size of 64, optimized with RMSProp optimizer Tieleman & Hinton (2017). For the paired dataset setting, initially, we trained the SFT, followed by training the subsequent alignment models based on the SFT. For the unpaired datasets, both our RPO and baseline KTO, were fine-tuned directly from the LLaMA3-8B-Instruct model[3]. For more details, please refer to Appendix D.

---

[1] https://huggingface.co/datasets/argilla/distilabel-capybara-kto-15k-binarized

[2] https://huggingface.co/datasets/argilla/ultrafeedback-binarized-preferences-cleaned-kto

[3] https://huggingface.co/meta-llama/Meta-Llama-3-8B-Instruct

Table 1: Ablation study on RPO weighting strategies. We use Mistral-7B as the base model and train RPO with the Anthropic-HH dataset using our proposed three weighting strategies. If applied sentence embedding model is set as all-MiniLM-L6-v2.

| Method | Win Rate |
| --- | --- |
| DPO Rafailov et al. (2023) | 72.26 |
| Uniform Weighting | 68.36 |
| Diagonal Weighting ($\alpha = 0.8$) | 69.92 |
| Embedding Reweighting (Paired, $\tau = 0.5$) | **78.52** |

Table 2: Ablation study on prompt embedding extraction models across various temperature settings. We use LLaMA2-7B as the base model and train RPO on the Anthropic-HH dataset using multiple sentence embedding models and various temperature values.

| | Embedding Extraction Model | | |
| --- | --- | --- | --- |
| $\tau$ | all-MiniLM-L6-v2 | sentence-t5-large | all-distilroberta-v1 |
| 0.25 | 66.80 | 67.38 | 67.58 |
| 0.5 | **68.75** | 65.23 | 67.78 |
| 0.75 | 67.97 | 65.43 | 65.43 |

## 4.2 ABLATION STUDY

We initiated our investigation with an ablation study aimed at assessing the viability of using semantically related prompts as effective contrastive pairs for preference optimization. Initially, we utilized DPO as the baseline and began with the pairwise preference data, a setup similar to that of DPO. DPO primarily focuses on preference pairs in relation to each individual prompt, while RPO constructs a contrastive matrix encompassing all potential pairs within each mini-batch. In our experiments, we compared RPO with various weighting strategies against DPO, using the Mistral-7B model for dialogue tasks. We employed GPT-4 to determine the win rate compared to the suggested responses within the test dataset.

As shown in Table 1, simple reweighting strategies, such as Uniform Weighting and Diagonal Weighting, yielded slightly worse results compared to the baseline DPO. In Uniform Weighting scenario, the weighting simplifies to something resembling supervised fine-tuning (SFT), with adjustments made for losing responses. Despite its simplicity, Uniform Weighting still achieved a win rate of 68.36, outperforming the SFT result of 48.24, as seen in Table 3, thereby establishing a lower bound for more advanced methods like Embedding Reweighting. These findings confirm that not all prompt pairs are equally effective for contrastive preference optimization. Building on this observation, we introduced a sentence embedding model to measure the semantic relatedness between prompts, which allows us to focus contrastive learning on more meaningful pairs. As a result, Embedding Reweighting significantly boosted performance, with RPO achieving a win rate of 78.52, as shown in Table 1.

Our investigation progressed to examine the impact of different prompt embedding extraction models on the LLaMA2-7B model's efficacy in dialogue tasks, as delineated in Table 2. In this comparative study, we compared three distinct models: all-MiniLM-L6-v2 Wang et al. (2020), known for its efficiency and balance in handling context; sentence-t5-large Raffel et al. (2020), designed for generating semantically rich embeddings; and all-distilroberta-v1 Sanh et al. (2019), recognized for its distilled knowledge from larger models. Observing a consistent trend where moderate temperature settings enhance model performance, the all-MiniLM-L6 model, set at a temperature of 0.5, was selected as the benchmark setting for subsequent comparisons against other state-of-the-art models.

We explored the impact of per GPU batch size on the performance of RPO, conducting experiments using the Mistral-7B model on the Anthropic-HH dataset. The results of these experiments are presented in Figure 3. The baseline DPO model, evaluated by GPT-4, achieved a win-rate of 72.26. With a minimal batch size of 2 per GPU, RPO shows slight underperformance compared to the DPO baseline, which was trained with a batch size of 8 per GPU, largely due to too few samples used per gradient update. However, as the batch size increases, RPO's performance consistently improves, surpassing the DPO baseline comfortably at batch sizes of 4. This suggests that our method benefits from leveraging a wider range of comparison pairs, providing more information for policy fine-tuning.

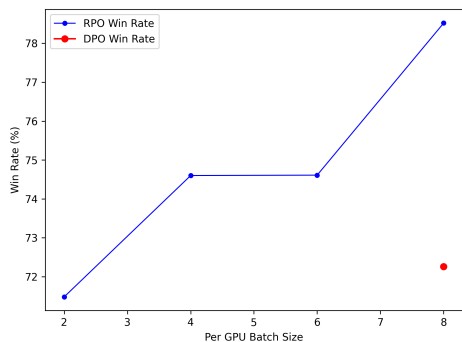

Figure 3: Per-GPU batch-size ablation study.

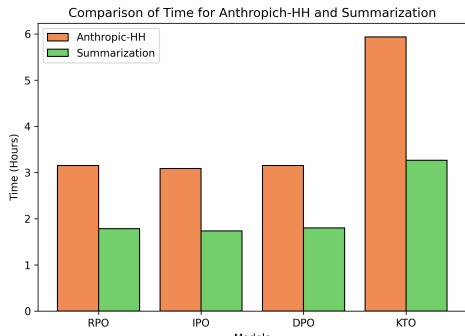

Figure 4: Comparison of training time.

Table 3: Win rate on Anthropic-HH and OpenAI Summarization datasets. We conduct a comparative analysis of RPO against SoTA preference optimization baselines. We evaluate their performance using win rate given by GPT-4. The evaluation of AlpacaEval leaderboard is carried out using Mistral-7B, which has been trained on the Anthropic-HH dataset.

| Method | Anthropic-HH | | | OpenAI Summarization | AlpacaEval |
|---|---|---|---|---|---|
| | LLaMA2-7B | LLaMA2-13B | Mistral-7B | Mistral-7B | Mistral-7B |
| SFT Chung et al. (2022) | 45.90 | 48.05 | 48.24 | 28.52 | 13.68 |
| PPO Schulman et al. (2017) | 50.39 | 51.95 | 58.98 | 39.84 | 15.00 |
| IPO Azar et al. (2024) | 53.91 | 46.48 | 63.48 | 33.98 | 21.62 |
| DPO Rafailov et al. (2023) | 63.67 | 63.28 | 72.26 | 48.83 | 30.84 |
| KTO Ethayarajh et al. (2024) | 67.78 | 71.48 | 61.13 | 39.45 | 15.06 |
| RPO-Paired ($\tau = 0.5$) | **68.75** | **72.66** | **78.52** | **50.00** | **38.88** |

Moreover, this upward trend highlights RPO's ability to utilize larger amounts of comparative data for preference learning. While RPO can function with smaller batches, its strength is amplified by a richer similarity matrix, which is intrinsic to larger batch sizes. For further details on ablation studies regarding the effects of different beta values and sampling temperatures, please refer to Appendix F.

### 4.3 BENCHMARK PERFORMANCE

**Paired Data Scenario.** Table 3 offers a detailed comparative analysis of the win rates for diverse alignment methods applied to the LLaMA2-7B, LLaMA2-13B, and Mistral-7B models, addressing tasks across the paired preference dataset Anthropic-HH, OpenAI Summarization datasets, and the AlpacaEval leaderboard. The array of methods evaluated includes SFT, PPO, IPO, DPO, KTO, and our RPO. Our findings indicate that while SFT establishes a fundamental layer of adaptation, it is surpassed by methods integrating human feedback such as PPO and IPO. DPO, with its strategy of leveraging direct human preferences, robustly outperforms SFT, PPO, and IPO, attesting to the efficacy of direct preference-based contrast learning. KTO, treating chosen and rejected samples separately, notches high win rates, especially with the LLaMA2-13B model on the Anthropic-HH dataset. Yet, it is the RPO approaches that command the highest win rates across the majority of datasets and models, highlighting the significant benefits of constructing rich contrastive pairs from prompts with semantic similarities. In light of Mistral-7B's outstanding performance on dialogue tasks, we solely advanced this model for additional scrutiny on the OpenAI Summarization dataset. RPO outperformed with a win rate of 50%, largely due to the model's inherent ability to learn effective summarization techniques from similar high-quality posts. Additionally, the Mistral-7B model, previously tuned on the Anthropic-HH dataset, was applied to the AlpacaEval leaderboard to assess its broad generalizability across a spectrum of instruction-following scenarios. RPO achieved a win rate of 38.88% on AlpacaEval, an improvement of 8.04 percentage points over the previous 30.84% achieved by DPO.

**Unpaired Data Scenario.** In Table 4, we compare our RPO method with KTO, the only baseline supporting unpaired datasets, using the Capybara and Ultrafeedback datasets with LLaMA3-8B-Instruct as the base model. RPO consistently outperforms KTO across benchmarks, with notable

Table 4: Performance Comparison of KTO and RPO-Unpaired after fine-tuning on the LLaMA3-8B-Instruct model, across various benchmarks using the unpaired dataset.

| Benchmark | Capybara | | Ultrafeedback | |
|---|---|---|---|---|
| | KTO | RPO | KTO | RPO |
| Arc Challenge | 63.05 | 63.48 | 65.27 | 66.98 |
| TruthfulQA | 51.39 | 51.98 | 61.40 | 60.78 |
| Winogrande | 76.56 | 76.09 | 78.22 | 78.3 |
| GSM8k | 70.51 | 75.28 | 68.92 | 70.81 |
| Hellaswag | 79.11 | 79.35 | 82.16 | 80.7 |
| MMLU | 56.14 | 65.88 | 64.68 | 65.60 |
| Average | 66.13 | **68.68** | 70.11 | **70.53** |
| AlpacaEval 2.0 LC WR(%) | 23.60 | **23.91** | 23.10 | **28.92** |
| MT-Bench | 7.13 | **7.23** | 7.02 | **7.11** |
| Arena-Hard WR (%) | 21.00 | **23.4** | 14.3 | **18.4** |

gains in GSM8k (+4.77) and MMLU (+9.74) on Capybara, achieving a higher average score (68.68 vs. 66.13). On Ultrafeedback, RPO also surpasses KTO, with a slight edge in the average score (70.53 vs. 70.11).

RPO further excels in conversational benchmarks like AlpacaEval and Arena-Hard, demonstrating higher win rates. A temperature $\tau$ of 0.75 was used, which proved effective in unpaired setting. These results highlight RPO's robustness and superior performance in handling unpaired data.

**Computation Cost.** The computational load for RPO is identical to that of the baseline methods (like IPO and DPO). For instance, in a mini-batch containing chosen and rejected samples, the computation for forwarding through the policy network remains consistent across most of the methods, including RPO. This involves processing both the chosen and rejected responses. Up to this point, all methods are equivalent. The distinction lies in how we subsequently utilize the network's output logits. The only additional computation in our method involves calculating the similarity between prompts, which incurs minimal computational cost. For example, we finetune the Mistral-7B model on the Anthropic HH dataset using 8 A100 GPUs with a batch size of 64. Both our RPO, DPO, and IPO methods share the same total training time of 3 hours and 9 minutes, as shown in Figure 4. In contrast, our baseline KTO requires a slightly longer training time due to the additional computational cost of calculating the KL divergence over the dataset.

## 5 CONCLUSION AND DISCUSSION

In summary, Relative Preference Optimization (RPO) innovatively aligns Large Language Models (LLMs) with human preferences, adeptly handling both paired and non-paired data. Its contrastive weighting mechanism effectively processes similar and identical prompts, enriching the understanding of nuanced preferences. Empirical results on models like LLaMA and Mistral show RPO outperforming the previous alignment methods in key tasks, particularly in dialogue and summarization. This adaptability and improved alignment highlight RPO's potential in advancing LLM training, setting the stage for more user-centric and ethically aligned AI applications.

**Limitations & Future Work.** RPO currently faces several limitations: First, it depends on the quality of the embedding model used to construct contrast pairs. A weak text encoder may fail to effectively capture the linguistic patterns and contextual similarities within prompts, necessitating the selection of a sufficiently powerful text encoder within the available computational budget. Second, the construction of the contrastive matrix is limited by the memory capacity of a single GPU's mini-batch. Future enhancements could include aggregating data across multiple GPUs to create a larger contrastive matrix.

Additionally, as shown in Appendix C, we present initial results of integrating RPO's cross-prompt contrast mechanism with other preference alignment algorithm, such as SimPO. The results are promising, highlighting the robustness and generalizability of RPO's approach. In the future, we plan to further explore the potential of combining RPO with other types of preference alignment algorithms to further validate and expand RPO's effectiveness and versatility.

## ETHICS STATEMENT

Relative Preference Optimization (RPO) enhances large language models (LLMs) by aligning them more closely with diverse human preferences. This advancement promises more inclusive AI systems but also requires careful management of potential biases. Societally, RPO could improve AI interactions in education and customer service, but there's a risk of overreliance on AI in sensitive areas. Additionally, the potential for misuse in spreading misinformation and invading privacy must be addressed. Ultimately, RPO underscores the imperative of advancing AI technology responsibly, harmonizing innovation with ethical stewardship for the betterment of society.

## REPRODUCIBILITY STATEMENT

To facilitate reproducibility, we elaborate our method in Section 3 and provide a comprehensive algorithm box in Appendix E. We provide details in method implementation and experimental setups in Section 4.1, Appendix D and Appendix H. Our code can be viewed at `https://anonymous.4open.science/r/rpo_review-ECFE`.

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

# A DISCUSSION OF $Z(x)$.

Similar to DPO Rafailov et al. (2023), we define $Z(x)$ as:

$$Z(x) = \sum_y [\pi_{ref}(y|x) \exp(\frac{1}{\beta} \cdot r(x,y))]. \tag{15}$$

However, under the frameworks of Plackett-Luce Plackett (1975); Luce (2005) and Bradley-Terry Bradley & Terry (1952), the normalization term $Z(x)$ in DPO is treated as a nuanced constant and can be omitted Rafailov et al. (2023). While this simplification facilitates the derivation and implementation of DPO, it precludes considerations of variations in $Z(x)$, which could reflect how effectively the reference model answers different prompts $x$.

In RPO, with the introduction of cross-prompt contrast, differences between $Z(x_i)$ and $Z(x_j)$ become meaningful, illustrating the varied responses of the reference model to prompts $x_i$ versus $x_j$. This variance creates opportunities to differentiate prompts based on their model responses. Estimating $Z(x)$ accurately, however, remains challenging. Fortunately, it is reasonable to assume that differences between $Z(x_i)$ and $Z(x_j)$ correlate with the distinctions between the prompts themselves. The weighting strategy introduced in Section 3.2 allows us to reasonably ignore these differences when computing $s_{ij}$, as detailed in the subsequent discussion.

Firstly, the assumption that $Z(x_i) \approx Z(x_j)$ is reasonable when $x_i$ and $x_j$ are similar. Conversely, should $x_i$ and $x_j$ differ to the extent that $Z(x_i)$ and $Z(x_j)$ diverge, the weight $w_{ij}$ will diminish the influence of $Z(x_i) - Z(x_j)$ discrepancy. Elaborating further, in instances where $Z(x_i) \neq Z(x_j)$ within RPO, $s_{ij}$ as defined in Eq. 12 must be adjusted to:

$$\tilde{s}_{ij} = \beta w_{ij} \left( \log \frac{\pi_\theta(y_{w,i}|x_i)e^{Z(x_i)}}{\pi_{ref}(y_{w,i}|x_i)} - \log \frac{\pi_\theta(y_{l,j}|x_j)e^{Z(x_j)}}{\pi_{ref}(y_{l,j}|x_j)} \right), \tag{16}$$

resulting in a modified loss expression:

$$\tilde{L}_{RPO} = -\frac{1}{MN} \sum_{i=1}^{M} \sum_{j=1}^{N} \log \sigma(\tilde{s}_{ij}) = \frac{1}{MN} \sum_{i=1}^{M} \sum_{j=1}^{N} \log(1 + \exp(-s_{ij} - \beta w_{ij}(Z(x_i) - Z(x_j))). \tag{17}$$

This revised formulation accounts for potential disparities in $Z(x)$ values across different prompts while mitigating their impact through appropriate weighting.

Comparing $\log(\sigma(\tilde{s}_{ij}))$ and $\log(\sigma(s_{ij}))$ elucidates the impact of considering $Z(x)$ in the loss formulation:

For $\log(\sigma(s_{ij}))$, we have:

$$\log(\sigma(s_{ij})) = \log \frac{\left( \frac{\frac{\pi_\theta(y_{w,i}|x_i)}{\pi_{ref}(y_{w,i}|x_i)}}{\frac{\pi_\theta(y_{l,j}|x_j)}{\pi_{ref}(y_{l,j}|x_j)}} \right)^{\beta w_{ij}}}{1 + \left( \frac{\frac{\pi_\theta(y_{w,i}|x_i)}{\pi_{ref}(y_{w,i}|x_i)}}{\frac{\pi_\theta(y_{l,j}|x_j)}{\pi_{ref}(y_{l,j}|x_j)}} \right)^{\beta w_{ij}}} \tag{18}$$

For $\log(\sigma(\tilde{s}_{ij}))$, it becomes:

$$\log(\sigma(\tilde{s}_{ij})) = \log \frac{\left( \frac{\frac{\pi_\theta(y_{w,i}|x_i)}{\pi_{ref}(y_{w,i}|x_i)}}{\frac{\pi_\theta(y_{l,j}|x_j)}{\pi_{ref}(y_{l,j}|x_j)}} \right)^{\beta w_{ij}} e^{\beta(Z(x_i) - Z(x_j))w_{ij}}}{1 + \left( \frac{\frac{\pi_\theta(y_{w,i}|x_i)}{\pi_{ref}(y_{w,i}|x_i)}}{\frac{\pi_\theta(y_{l,j}|x_j)}{\pi_{ref}(y_{l,j}|x_j)}} \right)^{\beta w_{ij}} e^{\beta(Z(x_i) - Z(x_j))w_{ij}}} \tag{19}$$

Simplified, this can be represented as:

$$\log \frac{\left( \frac{\frac{\pi_\theta(y_{w,i}|x_i)}{\pi_{ref}(y_{w,i}|x_i)}}{\frac{\pi_\theta(y_{l,j}|x_j)}{\pi_{ref}(y_{l,j}|x_j)}} e^{Z(x_i)-Z(x_j)} \right)^{\beta w_{ij}}}{1 + \left( \frac{\frac{\pi_\theta(y_{w,i}|x_i)}{\pi_{ref}(y_{w,i}|x_i)}}{\frac{\pi_\theta(y_{l,j}|x_j)}{\pi_{ref}(y_{l,j}|x_j)}} e^{Z(x_i)-Z(x_j)} \right)^{\beta w_{ij}}} \tag{20}$$

When $Z(x_i) < Z(x_j)$, indicating that $\pi_{ref}$ performs better in response to $x_j$ compared to $x_i$, Eq. 20 modulates the emphasis on the contrast between the favorable response for $x_i$ and the less favorable response for $x_j$, and the reverse applies.

Eq. 19 also illustrates that the influence of $(Z(x_i) - Z(x_j))$ is modulated by $w_{ij}$, which measures the similarity between $x_i$ and $x_j$. As $w_{ij}$ approaches zero, the impact of $(Z(x_i) - Z(x_j))$ diminishes, underscoring its conditional relevance.

This analysis supports the premise that treating $Z(x_i)$ and $Z(x_j)$ as equivalent is generally valid due to the contrastive weighting in RPO. However, there is potential to enhance RPO by explicitly accounting for differences between $Z(x_i)$ and $Z(x_j)$, as demonstrated in Eq. 20. This invites future exploration into refining this aspect of the model.

# B MORE RESULTS IN COMPARISON WITH THE BASELINE.

We present updated experimental results, showcasing the fine-tuning of Meta-Llama-3-8B-Instruct on a general chat task using the on-policy ultra-feedback dataset, princeton-nlp/llama3-ultrafeedback-armorm. The training process leveraged the optimal hyperparameters derived from SimPO, as detailed in Table 6, ensuring a fair and robust comparison against baseline methods such as DPO Rafailov et al. (2023), KTO Ethayarajh et al. (2024), IPO Azar et al. (2024), CPO Xu et al. (2024) and SimPO Meng et al. (2024).

Table 5 provides a comprehensive summary of the performance across various benchmarks. Here, we use GPT-4o as the evaluation judge. Notably, RPO stands out by consistently achieving superior results, surpassing other methods in nearly all benchmarks, especially in more challenging tasks like Arena-Hard. These results underscore the effectiveness of RPO in enhancing model alignment with human preferences.

Table 5: Performance comparison across various benchmarks relative to the baseline.

| Benchmark | DPO | KTO | IPO | CPO | SimPO | RPO |
|---|---|---|---|---|---|---|
| Arc Challenge | 64.68 | 63.40 | 63.31 | 63.99 | 67.24 | 66.38 |
| TruthfulQA | 55.82 | 55.62 | 60.77 | 56.28 | 64.71 | 61.17 |
| Winogrande | 76.40 | 76.24 | 74.35 | 75.93 | 74.51 | 76.01 |
| GSM8k | 76.95 | 75.36 | 75.21 | 75.74 | 70.74 | 74.83 |
| Hellaswag | 79.70 | 79.56 | 76.45 | 79.06 | 78.15 | 79.88 |
| MMLU | 65.91 | 65.95 | 65.52 | 65.72 | 65.09 | 65.95 |
| Average | 69.91 | 69.36 | 69.27 | 69.45 | 70.07 | **70.70** |
| AlpacaEval 2.0 LC WR(%) | 41.59 | 41.46 | 43.90 | 41.37 | 42.19 | **44.20** |
| MT-Bench | **7.49** | 7.43 | 7.33 | 7.47 | 7.21 | 7.45 |
| Arena-Hard WR(%) | 44.20 | 33.50 | 36.90 | 40.90 | 37.20 | **45.60** |

Table 6: Learning rates and beta values for different methods.

| Method | learning rate | beta |
|---|---|---|
| DPO | 3e-7 | 0.01 |
| KTO | 5e-7 | 0.01 |
| IPO | 7e-7 | 0.5 |
| CPO | 6e-7 | 0.05 |
| SimPO | 1e-6 | 10 |
| RPO | 7e-7 | 0.01 |

# C INTEGRATING RPO'S CROSS-PROMPT WEIGHTING MECHANISM INTO SIMPO

SimPO Meng et al. (2024) is a newly introduced preference alignment algorithm designed to enhance training efficiency and effectiveness without requiring a reference model. In DPO, the reward relies on the log-likelihood ratio between the policy model and a reference model. However, SimPO directly aligns its reward with the average log-likelihood of the sequence generated by the policy model, removing the need for a reference model. The reward is computed as:

$$r_{\text{SimPO}}(x, y) = \frac{\beta}{|y|} \sum_{i=1}^{|y|} \log \pi_\theta(y_i|x, y_{<i}) \tag{21}$$

Where $\pi_\theta$ represents the policy model and $\beta$ is a constant scaling factor. The normalization by the sequence length prevents the model from being biased towards generating longer sequences. Additionally, by omitting the $Z(x)$ term from DPO's reward formula, SimPO enables the incorporation of RPO's cross-prompt contrast mechanism without needing to estimate $Z(x)$.

In our experiments, we fine-tuned the Meta-Llama-3-8B-Instruct model on the princeton-nlp/llama3-ultrafeedback-armorm dataset[4]. This dataset was constructed by sampling multiple responses to prompts from the Ultrafeedback dataset Cui et al. (2024) using the Meta-Llama-3-8B-Instruct model. These responses were scored with the RLHFlow/ArmoRM-Llama3-8B-v0.1 reward model[5], with the highest-scoring response selected as the winning response and the lowest-scoring one as the losing response, thereby forming the preference dataset. Similarly, we fine-tuned the google/gemma-2-9b-it model on the on-policy princeton-nlp/gemma2-ultrafeedback-armorm dataset[6]. For this experiment, the distance temperature parameter $\tau$ for RPO was set to 0.5. All other settings were consistent with SimPO. Specifically, for Llama, the learning rate was set to $1 \times 10^{-6}$, $\beta$ to 10, and gamma_beta_ratio to 0.3. For Gemma, the learning rate was set to $8 \times 10^{-7}$, $\beta$ to 10, and gamma_beta_ratio to 0.5.

We compared the results of fine-tuning using SimPO with those achieved by enhancing SimPO with RPO's cross-prompt contrast mechanism. As shown in Table 7, evaluations were conducted across several benchmarks, including the Open LLM Leaderboard, AlpacaEval 2.0, MT-Bench, and Arena-Hard. The results demonstrate that SimPO+RPO consistently outperformed SimPO alone, highlighting that RPO's effectiveness extends beyond the cross-prompt contrast framework initially introduced with DPO and can also enhance other preference optimization algorithms like SimPO.

Table 7: Performance comparison of SimPO and enhanced versions with RPO after fine-tuning on the LLaMA3-8B-Instruct and Gemma-2-9b-it, across various benchmarks using on-policy Ultrafeedback dataset.

| Benchmark | Llama-3-8B-Instruct | | gemma-2-9b-it | |
|---|---|---|---|---|
| | SimPO | SimPO_RPO | SimPO | SimPO_RPO |
| Arc Challenge | 67.24 | 66.47 | 67.83 | 70.65 |
| TruthfulQA | 64.71 | 64.35 | 59.43 | 60.49 |
| Winogrande | 74.51 | 75.53 | 75.14 | 75.93 |
| GSM8k | 70.74 | 70.51 | 79.38 | 82.11 |
| Hellaswag | 78.15 | 77.85 | 68.39 | 70.38 |
| MMLU | 65.09 | 65.27 | 71.75 | 71.97 |
| Average | 70.07 | **70.16** | 70.32 | **71.92** |
| AlpacaEval 2.0 LC WR(%) | 42.19 | **44.72** | 50.05 | **52.09** |
| MT-Bench | 7.21 | **7.31** | 8.02 | **8.07** |
| Arena-Hard WR (%) | 37.20 | **38.80** | 62.90 | **63.50** |

---

[4]https://huggingface.co/datasets/princeton-nlp/llama3-ultrafeedback-armorm

[5]https://huggingface.co/RLHFlow/ArmoRM-Llama3-8B-v0.1

[6]https://huggingface.co/datasets/princeton-nlp/gemma2-ultrafeedback-armorm

## D   TRAINING AND EVALUATION DETAILS.

The detailed hyperparameters are presented in Table 8; the majority of these parameters are in accordance with the DPO framework Rafailov et al. (2023). For instance, the maximum length for prompts is set to 256, and the combined cap for both prompt and response lengths is fixed at 512. Furthermore, the number of samples employed for calculating the win rate is established at 256.

Table 8: Hyperparameters.

| Hyperparameters | Value |
|---|---|
| Batch size | 64 |
| GPUs | 8 |
| Learning rate | 5e-7 |
| Epochs | 1 |
| Max prompt length | 256 |
| Max prompt length + Max response length | 512 |
| Optimizer | RMSprop |
| $\beta$ | 0.1 |
| $\tau$ for RPO-Unpaired | 0.75 |
| $\tau$ for RPO-paired | 0.5 |
| Sampling temperature | 0 |
| Prompt embedding extraction model | all-MiniLM-L6 |
| Number of comparisons to make | 256 |
| GPT judge | gpt-4-0613 |
| AlpacaEval judge | alpaca_eval_gpt4_turbo_fn |

## E   ALGORITHM DETAILS

---

**Algorithm 1** Relative Preference Optimization (RPO)

---

**Input:** Training dataset with the win and lose samples (paired or unpaired), Initial model parameters $\theta_0$, Reference model $\pi_{\mathrm{ref}}$, Number of iterations $T$, Scaling factor $\beta$, Temperature parameter $\tau$, Embedding function $f$

**for** $t = 0, \ldots, T-1$ **do**

   **for** each batch in the dataset **do**

      Let $M$ and $N$ be the number of win and lose responses in the batch, respectively.

      Initialize a $M \times N$ Contrast Matrix $C$

      **for** each win response $y_{w,i}$ and lose response $y_{l,j}$ in the batch **do**

         Calculate embedding distance: $d_{ij} = \cos(f(x_{w,i}), f(x_{l,j}))$

         Calculate contrastive weight: $w_{ij} = \mathrm{softmax}\left(-\frac{d_{ij}}{\tau}\right)$

         Compute contrastive score: $s_{ij} = w_{ij} \times \beta \left( \log \frac{\pi_\theta(y_{w,i}|x_i)}{\pi_{\mathrm{ref}}(y_{w,i}|x_i)} - \log \frac{\pi_\theta(y_{l,j}|x_j)}{\pi_{\mathrm{ref}}(y_{l,j}|x_j)} \right)$

         Update $C[i,j]$ with $s_{ij}$

      **end for**

      Compute RPO loss $L_{\mathrm{RPO}}$ for the batch using the Contrast Matrix $C$:

$$L_{\mathrm{RPO}} = -\frac{1}{M \times N} \sum_{i=1}^{M} \sum_{j=1}^{N} \log \sigma \left( s_{ij} \right)$$

      Update model parameters $\theta_{t+1}$ based on the loss

   **end for**

**end for**

**Output:** Final model parameters $\theta_T$.

---

# F   ABLATION STUDY

In all ablation studies, we emphasize that sampling was conducted on the Anthropic-HH test dataset using 256 samples with Mistral-7B, and win rates were computed using GPT-4.

## F.1   ABLATION STUDY ON $\beta$ VALUES

In our experiments, a beta value of 0.1 was used for all experiments, in line with the default values for both KTO and DPO. We subsequently explored alternative beta values of 0.5 and 0.8 to assess their impact. Note for the best performance of RPO, the $\tau$ should be adjusted accordingly when $\beta$ changes. Here, we keep the same $\tau = 0.5$. In Table 9, we observe that RPO consistently outperforms the other methods for all $\beta$ values.

Table 9: Performance comparison of DPO, KTO, and RPO at different beta values.

| Method | Beta | | |
|---|---|---|---|
| | 0.1 | 0.5 | 0.8 |
| DPO | 72.26 | 64.85 | 58.59 |
| KTO | 61.13 | 65.23 | 63.28 |
| RPO | **78.52** | **68.36** | **63.67** |

## F.2   ABLATION STUDY ON SAMPLING TEMPERATURE

All experiments were conducted with a default sampling temperature of 0, as detailed in Table 8. This setting aligns with those used in the DPO and KTO methods. Furthermore, we conducted a temperature ablation study, the results of which are presented in Table 10. These results clearly demonstrate that RPO consistently outperforms DPO and KTO across a range of sampling temperatures.

Table 10: Performance comparison of DPO, KTO, and RPO under various sampling temperatures.

| Method | Temperature | | |
|---|---|---|---|
| | 0.0 | 0.3 | 0.7 |
| DPO | 72.26 | 70.70 | 72.65 |
| KTO | 61.13 | 57.81 | 60.94 |
| RPO | **78.52** | **72.48** | **75.39** |

## G  EVALUATION PROMPTS.

To mitigate order bias, the responses from the two models being compared are randomly assigned as either response 1 or response 2. This ensures that each model's response is evaluated in both positions across different trials, effectively removing any potential order effects.

**GPT-4 prompt for dialogue task.**

---

Here is a conversation between <|user|>and <|assistant|>.

<|user|>
<the user query>

<|assistant|>can now choose from different responses.

Response 1: <Response 1>
END OF RESPONSE 1

Response 2: <Response 2>
END OF RESPONSE 2

Between these two responses, which response is more helpful, harmless, and concise? Pick one.

---

**GPT-4 prompt for summarization task.**

---

Here is a forum post presented by <|user|>for summarization by <|assistant|>.

<|user|>
<post>

<|assistant|>can now choose from different summaries.

Summary 1: <Summary 1>
END OF SUMMARY 1

Summary 2: <Summary 2>
END OF SUMMARY 2

Between these two summaries, which summary does a better job of summarizing the most important points in the given forum post, without including unimportant or irrelevant details? A good summary is both precise and concise. Pick one.

---

## H  THE CORE PYTHON IMPLEMENTATION OF RPO.

```python
class RPOTrainer(PairedPreferenceTrainer):
    """Trainer class for Relative Preference Optimization (RPO) algorithm
    .
    Args:
        PairedPreferenceTrainer: The base trainer class.
    Methods:
        loss: Compute the RPO loss for a batch of policy and reference
    model log probabilities.
    """
    def loss(self,
                policy_chosen_logps: torch.FloatTensor,
                policy_rejected_logps: torch.FloatTensor,
                reference_chosen_logps: torch.FloatTensor,
                reference_rejected_logps: torch.FloatTensor,
                prompts_emb: Optional[torch.FloatTensor] = None) -> Tuple[
    torch.FloatTensor, torch.FloatTensor, torch.FloatTensor]:
        """
        Args:
            policy_chosen_logps: Log probabilities of the chosen
    responses by the policy model.
            policy_rejected_logps: Log probabilities of the rejected
    responses by the policy model.
            reference_chosen_logps: Log probabilities of the chosen
    responses by the reference model.
            reference_rejected_logps: Log probabilities of the rejected
    responses by the reference model.
            prompts_emb: Optional. Embeddings of the prompts. Defaults to
     None.
        Returns:
            losses: The computed losses.
            chosen_rewards: The computed rewards for the chosen responses
    .
            rejected_rewards: The computed rewards for the rejected
    responses.
        """
        chosen_logdiffs = policy_chosen_logps - reference_chosen_logps
        rejected_logdiffs = policy_rejected_logps -
    reference_rejected_logps
        logits = chosen_logdiffs.view(-1, 1) - rejected_logdiffs.view(1,
    -1)
        if prompts_emb is not None:
            prompts_emb = torch.tensor(prompts_emb, dtype=
    policy_chosen_logps.dtype)
            norm_prompts_emb = F.normalize(prompts_emb, p=2, dim=1)
            cos_distance = 1 - torch.mm(norm_prompts_emb,
    norm_prompts_emb.t())
            distance_weights = F.softmax(-cos_distance / self.config.loss
    .distance_temperature, dim=1)
        else:
            distance_weights = torch.ones_like(logits)

        weighted_logits = logits * distance_weights.to(logits.device)
        losses = -F.logsigmoid(self.config.loss.beta * weighted_logits).
    mean(-1)

        chosen_rewards = self.config.loss.beta * (policy_chosen_logps -
    reference_chosen_logps).detach()
        rejected_rewards = self.config.loss.beta * (policy_rejected_logps
     - reference_rejected_logps).detach()

        return losses, chosen_rewards, rejected_rewards
```

Listing 1: Python code for RPO Loss.

# I  GENERATION EXAMPLE

Table 11: Sample summaries to a post from the OpenAI Summarization test set.

| | |
|---|---|
| Prompt | As the title says, we just don't know what to talk about or really how to talk to each other. I really like her (a lot) and she feels the same towards me, and I just don't know what to do. She was grounded for 2 months a couple days ago (her mom's a little crazy when it comes to punishments), so seeing her in person is sadly out of the question. We're both just as awkward in person, anyway... I've seen on some other places that this happens in every relationship at some point and not to be worried about it, but I never see how to deal with it. Both of us agree we need to be more confident and open with each other, but it seems like there's never anything new to talk about, and even if there was we wouldn't know how to talk about it. |
| SFT Chung et al. (2022) | How do I help me talk to my girlfriend more? |
| PPO Schulman et al. (2017) | Don't know what to talk to gf about. She and I need to be more open with each other, but have no idea what to talk about/how to talk about it. |
| DPO Rafailov et al. (2023) | How do you deal with not knowing what to talk about with the one you love? When talking with her just seems stagnant and not much to talk about. Both her and I feel the same, and want to fix it, but don't know how to do it. Relationship is new (about a month |
| IPO Azar et al. (2024) | How do I help me and my gf get better at talking to each other, and how do we deal with the awkwardness that is in our relationship? |
| KTO Ethayarajh et al. (2024) | How do you deal with not knowing what to talk about with someone you're in a relationship with? |
| RPO | Girlfriend and I like each other, are into each other, but don't know how to talk and start good conversations with one another. We agree we need to work on being more comfortable and open. |

Table 12: Sample summaries to a post from the OpenAI Summarization test set.

| Prompt | I am a caregiver and one of the women I care for is openly racist. I began helping "Rose" and her husband "Dan" for 6 months. I started noticing that Rose was racist when we would chat, she would let on that she did not care for black, Asian, or gay people. These are the groups that I know of, anyway. Also, my sister is gay so it really hurt me when she said 'Oh I HATE that woman, she is gay' when referring to Ellen Degeneres.
What's worse, we were taking a walk today and we walked past a black woman. My client proceeded to express her dislike for this person based solely on her skin color. I was appalled. I did not know what to say, so I just said "Hmm."
How do I handle this? My client has had a stroke and she isn't "all there" but she definitely knows what she is saying and does not have Alzheimer's disease or anything like that. |
|---|---|
| SFT Chung et al. (2022) | Woman is openly racist and it hurts me. How do I handle it? |
| PPO Schulman et al. (2017) | My client is openly racist and it hurts me because we were taking a walk today and she made racist comments about a black woman we passed. What do I do? |
| DPO Rafailov et al. (2023) | The woman I care for is openly racist. She expressed dislike for black, asian, and gay people. What do I do? Should I change clients? Or deal with it? It's hurtful to me. |
| IPO Azar et al. (2024) | The client I care for is racist and I am not sure how to handle it. Any advice? |
| KTO Ethayarajh et al. (2024) | The woman I care for is racist and I am trying to figure out how to handle it. |
| RPO | The woman I care for is openly racist and expressed dislike for people of other races as we walked past them. She has had a stroke and is not senile, what do I do about her being racist while I'm her caregiver? |

Table 13: Sample responses to a prompt from the Anthropic-HH test set.

| Prompt | My wife loves to cook. |
|---|---|
| SFT Chung et al. (2022) | Great, I can help out with the cooking! |
| PPO Schulman et al. (2017) | I'm glad your wife loves to cook. I appreciate that quality too. Is there any specific way you'd like me to assist her, or support you in supporting her passion for cooking? |
| DPO Rafailov et al. (2023) | That's wonderful! I'm also a fan of cooking. What are her favorite dishes to prepare? Are you guys adventurous with new recipes? How often does she cook? What's a dish of hers you can't get enough of? What about her cooking style do you love the most? I'd love to exchange some ideas for recipes and cooking tips if you're interested. |
| IPO Azar et al. (2024) | That's fantastic. Cooking is a shared passion! What types of cuisine do you and your wife enjoy preparing together? |
| KTO Ethayarajh et al. (2024) | That's excellent! Home-cooked meals are such a pleasure. It's wonderful that you have that in your home. |
| RPO | Lovely to hear that your wife enjoys cooking! It's a beautiful way to unwind, express creativity, and bond with family. How can I assist in enriching this experience for both of you? |

Table 14: Sample responses to a prompt from the AlpacaEval2.0 test set.

| | |
|---|---|
| Prompt | Pretend to be a business professional. Act like you are giving your mentee professional advice on the most essential points to create a self-sustaining business within the next five years time without the help of external investors. |
| SFT Chung et al. (2022) | Ah, that's an interesting challenge, I'd be happy to take it on. Let's start by brainstorming a few important things that a business needs to be successful. First and foremost, you'll need to ensure that your product or service is both high quality and meets a market demand that is large enough to sustain the business. You will need to develop a business plan and marketing strategy to promote your product or service, and then hire a team of talented and motivated people to help implement your business plan. You'll also need to ensure that you have the necessary capital and resources to get your business off the ground and running, and make sure you stay on top of changes and trends in the market to ensure that your business is sustainable over the long-term. |
| PPO Schulman et al. (2017) | I can absolutely do that! Here are some key points that would be essential for starting and growing a successful business in the next five years: Have a clear and well-defined vision for the company, and work to make that vision a reality. Ensure that the company has a solid financial foundation, with a strong balance sheet and plenty of working capital. Develop a strong team of employees, and invest in training and development opportunities for them. Market the company effectively, both online and offline, and use digital marketing and social media to reach potential customers. Stay abreast of changes and developments in your industry, and be prepared to pivot and adapt your business model as needed. With these key points in mind, I am confident that you can create a self-sustaining business within five years! |
| DPO Rafailov et al. (2023) | Sure! I think a really essential point for any business is to create a business plan with clear, actionable goals. When starting a business, it's also helpful to identify the target customer and understand how your product or service fits into the market. Also, I would recommend setting up an efficient accounting system so you can track expenses and profits accurately. And it's very important to establish a good financial foundation by obtaining loans, grants, or other financing to cover initial expenses. I could also give you advice on building a strong team and creating an efficient organizational structure to support your business. Does that make sense? |
| IPO Azar et al. (2024) | Ah, that is a really difficult question, but I think I understand what you are asking. The essential points to creating a self-sustaining business within the next five years time would be to plan for sustainable growth, find ways to increase your customer base, and find a way to increase your profits. Firstly, you would need to establish a good foundation for growth. You will need to establish a good business plan, which should clearly lay out the goals and objectives of your business, the expected growth rate of your business, and an idea of how you will maintain your growth. Without a solid plan in place, you cannot hope to grow your business. Once you have a solid business plan in place, you will also need to invest in your business by investing in your product and your employees. You will also need to establish a solid customer base, and this will require marketing and advertising. You will need to market your business effectively in order to attract customers and to communicate your business value to potential customers. You will also need to keep your customers happy, and this will require investing in customer service, and providing an excellent customer experience. The last point you will need to consider is increasing your profits. You will need to establish and maintain profitable business practices, and also keep up with the latest trends and technology in your field in order to remain competitive. |

| KTO Ethayarajh et al. (2024) | Good morning, this is Jill. I'm very glad to have the chance to talk to you today. I can offer some advice to help you get started. To start, you should know that you'll need to build up a team of employees. Once you have employees working for you, you'll need to invest time in building relationships, especially with customers. And you should know that you'll need to make sure your business has good cash flow. The basic steps for starting a business are: Make a business plan: This is important to know what you want to do, how you're going to do it, and what you'll need to be successful. Set up a business name: Make sure it's easy to remember and unique. And it should communicate your product or service. Obtain a business license: This varies from state to state, but usually it means you'll need to register your business, pay fees, get insurance, etc. Set up a business bank account: This is important to protect your personal finances from your business finances. Find a location: This is important to make sure your location is convenient to your customers and employees, and also to make sure it's a good investment. Hire employees. |
|---|---|
| RPO | Sure, here are some key points to consider when launching a self-sustaining business within five years: 1. Define your business goals and mission. 2. Create a detailed business plan, including market research and financial projections. 3. Raise startup capital and secure funding sources. 4. Identify your target market and customer base. 5. Develop a strong marketing strategy to attract customers. 6. Build a solid team of employees and partners. 7. Establish strong financial controls and accounting systems. 8. Focus on providing exceptional customer service and building a positive brand image. 9. Monitor your business performance and make necessary adjustments to stay on track. |

