# OpenReview forum: "Relative Preference Optimization: Enhancing LLM Alignment through Contrasting Responses across Identical and Diverse Prompts"
_ICLR.cc/2025/Conference — Submitted to ICLR 2025_

### Official Review · Reviewer_i7yC · 2024-11-03

**Soundness:** 3
**Presentation:** 3
**Contribution:** 2
**Rating:** 5
**Confidence:** 4

**Summary:**

The paper introducs Relative Preference Optimization (RPO), a novel method to enhance the alignment of LLMs with human preferences. RPO considers preferences across both identical and semantically related prompts and employs a contrast matrix to discern preferences between preferred responses via a contrastive weighting mechanism. So RPO enhancs model training with a broader range of data. Empirical tests on dialogue and summarization tasks showed that RPO outperforms existing methods like DPO, IPO, and KTO, especially in adapting to varied contexts.

**Strengths:**

- Extends beyond traditional methods by considering semantically related prompts.
- RPO is adaptable to both paired and unpaired data scenarios, making it a flexible tool for different types of preference data.

**Weaknesses:**

- RPO relies heavily on the semantic similarity of prompts to establish contrastive pairs. This approach might limit its effectiveness in contexts where prompts lack clear semantic relationships, especially in diverse general-purpose task datasets. As a result, the probability of relevant prompts appearing within a mini-batch is relatively low, and it is rather difficult to screen out sample pairs of high $\omega$ within a mini-batch. Related to Question 1.
- As an article submitted to ICLR, both the scientific problems distilled in the article and its methodologies are rather deficient in terms of inspiration and theoretical contributions.

**Questions:**

1. It seems that the performance improvement of this method relative to DPO mainly stems from the combination of additional weighted training samples within the mini-batch, which expands the scale of the training data. Could you please count the proportion of samples with omega > 0.5 in the Anthropic-HH training set under different configurations of batch size?

2. If we construct new sample pairs of relative prompts (i.e. high $\omega$ values) based on the entire dataset and then add them to the original dataset according to the ratio obtained from Question 1, could DPO produce comparable or even better result?

---

> ### Author Response · Authors · 2024-11-22
> **Response to Reviewer i7yC**
>
> We thank Reviewer i7yC for providing detailed feedback and valuable suggestions. We provide further clarifications below.
>
> **W1:** Thank you for your question. In the context of paired data, we provide an analysis of a random mini-batch from the weight matrix with a per-GPU batch size of 8 and a distance temperature $\tau = 0.5$, as shown in Q1. The diagonal entries (corresponding to response pairs from the same prompt) exhibit weights fluctuating around 0.5, dominating the loss contribution. In contrast, weights for other samples fluctuate around 0.1, based on their similarity to the corresponding prompt. This indicates that even in a random mini-batch, the model effectively learns additional information from rejected responses to other prompts.
>
> Moreover, we have demonstrated the effectiveness of RPO on general-purpose datasets, such as ultrafeedback, through results presented in Table 1 and Table 2 (refer to the general rebuttal section). Additionally, in Table 4 of the manuscript, we showcase results under a completely random unpaired setting on a general chat task. Even in this setting, RPO outperforms KTO. While KTO fundamentally focuses on increasing the reward of chosen responses and reducing the reward of rejected responses, RPO further enhances this process by leveraging prompt similarity to assign differentiated loss weights to various pairs.
>
> In summary, we have validated the effectiveness of RPO across both general-purpose datasets (ultrafeedback/capybara) and specific-task datasets (HH/Summarization). The mini-batch similarity issue does not detract from RPO’s ability to effectively utilize rejected responses and outperform baseline methods, even in diverse datasets.
>
> **W2:**  We strongly disagree with the dismissive tone in your assessment of this paper's contributions. We submitted our work to ICLR because we are genuinely excited about sharing our insights with the community and believe in their value. We respectfully request that you also review the comments from other reviewers, which highlight the significance of our methodological contributions. Taking our response into account, we kindly ask you to consider adjusting your position.
>
> Our work addresses a critical gap in the original DPO framework, specifically the lack of comparisons across prompts. Inspired by human learning processes, we propose constructing a contrastive matrix to incorporate contrastive learning with relative prompt-answer pairs. Through extensive experiments, we have validated the effectiveness of this approach.
>
> Furthermore, the concept of Cross-Prompt Comparisons introduced in RPO extends beyond its current scope. For instance, it can be applied to reward model training. This idea also opens up opportunities for exploration in other areas of RLHF. We believe that simple yet effective methods like RPO are both valuable and capable of driving meaningful advancements.
>
> **Q1:**
> 1. **Proportion of weights > 0.5**:
>
> The proportion of weights greater than 0.5 under different per-GPU batch sizes (for the Anthropic HH dataset with $ \tau = 0.5$) is summarized as follows:
>
>    | **Per-GPU Batch Size** | **Proportion of Weights > 0.5** |
>    |-------------------------|--------------------------------|
>    | 2                       | 50.00%                       |
>    | 4                       | 24.90%                       |
>    | 6                       | 12.18%                       |
>    | 8                       | 0.55%                        |
>
> In our final experiments (with a per-GPU batch size of 8), the proportion of weights greater than 0.5 is only 0.55%.
>
> 2. **Examples from specific mini-batches**:
>    - Per-GPU batch size = 2:
>      ```
>      [[0.8516, 0.1484],
>       [0.1484, 0.8516]]
>      ```
>    - Per-GPU batch size = 4:
>      ```
>      [[0.6602, 0.1147, 0.0913, 0.1328],
>       [0.1206, 0.6953, 0.0845, 0.1001],
>       [0.0991, 0.0869, 0.7148, 0.0972],
>       [0.1357, 0.0972, 0.0913, 0.6758]]
>      ```
>    - Per-GPU batch size = 6:
>      ```
>      [[0.5156, 0.0898, 0.0713, 0.1040, 0.0986, 0.1206],
>       [0.1006, 0.5781, 0.0703, 0.0835, 0.0830, 0.0835],
>       [0.0815, 0.0713, 0.5859, 0.0796, 0.0879, 0.0928],
>       [0.1094, 0.0781, 0.0732, 0.5430, 0.1069, 0.0894],
>       [0.1030, 0.0771, 0.0811, 0.1064, 0.5391, 0.0933],
>       [0.1250, 0.0771, 0.0845, 0.0879, 0.0923, 0.5352]]
>      ```
>    - Per-GPU batch size = 8:
>      ```
>      [[0.4102, 0.0713, 0.0569, 0.0830, 0.0786, 0.0962, 0.1177, 0.0859],
>       [0.0864, 0.4980, 0.0605, 0.0718, 0.0713, 0.0718, 0.0767, 0.0635],
>       [0.0698, 0.0613, 0.5039, 0.0684, 0.0757, 0.0801, 0.0645, 0.0747],
>       [0.0845, 0.0605, 0.0569, 0.4199, 0.0825, 0.0693, 0.1455, 0.0806],
>       [0.0845, 0.0635, 0.0664, 0.0874, 0.4434, 0.0762, 0.0845, 0.0952],
>       [0.1060, 0.0654, 0.0718, 0.0747, 0.0781, 0.4531, 0.0820, 0.0684],
>       [0.1138, 0.0610, 0.0505, 0.1377, 0.0757, 0.0718, 0.3965, 0.0923],
>       [0.0923, 0.0559, 0.0649, 0.0845, 0.0942, 0.0664, 0.1021, 0.4395]]
>      ```

---

> ### Author Response · Authors · 2024-11-22
> **Response to Reviewer i7yC**
>
> **Q2:** Thanks for your question. The approach you suggest can indeed be regarded as an offline variant of RPO.
> In our experiments, we first constructed an 8-nearest HH dataset based on prompt similarity and then applied RPO to expand the original data with additional contrastive pairs for training. However, we found that the results of this offline approach were comparable to those achieved by our current online version.
>
> From a statistical perspective, both offline RPO, which uses a global similarity matrix, and online RPO, which leverages a batch-wise similarity matrix, are designed to achieve similar objectives. For instance, stochastic gradient descent often matches or even surpasses global gradient descent in performance. Furthermore, our results consistently show that RPO outperforms DPO, further validating our claim that introducing additional contrastive pairs enhances performance.

---

> ### Comment · Reviewer_i7yC · 2024-11-24
>
> Thanks for your detailed response. Through careful consideration regarding the useful information that can still be mined among samples within the training batch, I acknowledge that although the method described in the paper is not an excellent solution, I recognize the value in your observation of this phenomenon. Therefore, I have decided to adjust the score given.
>
> I share a similar view with Reviewer hL8X, believing that the main benefit of RPO comes from the runtime data augmentation, where the augmentation involves creating new pairs among similar prompts. Due to the high sparsity of text sequences, any method that increases data density in the text sequence space will be effective. This is also one of the important research directions for LLM at present. Let's consider a thought experiment: Suppose I have a dataset $S=\\{x_i,y_i\\}$, and construct a dataset $D=\\{x_i,y_w=y_i,y_l=y_j\\}, j \ne i$, where. Assuming the method of constructing the dataset is unknown, can we achieve an improvement in model performance through RPO?
>
> I apologize for an incorrect symbol reference in W1 and Q1. My intention was to express that "the probability of high similarity among prompts in a small batch is low." Therefore, I should have referred to the symbol $\tilde{\omega}$ at Eq. 10 instead of $\omega$ at Eq. 11. Based on this change and considering your response regarding Q1, I am concerned about the phenomenon when "Per-GPU batch size = 2," where a high-weight (high $\omega$) but potentially low-value (low $\tilde{\omega}$) training pair is always created. This implies that effective sample pairs can be derived from any two unrelated prompts. On the other hand, in the case of large batches, even if all prompts are highly related, the normalization effect of Eq. 11 causes originally high-value (high $\tilde{\omega}$) samples that could be formed through pairwise combinations to become low-weight (low $\omega$). These two examples illustrate the areas where RPO as a training optimization method may be underconsidered.
>
> So, to my personal opinion, it might be a better choice for this paper to be organized from the perspective of data augmentation rather than training optimization. I believe that the contribution level of a chosen-reject sample pair to the training task should be an objective constant, meaning it should be more related to $\tilde{\omega}$ rather than to batch size or other samples within the batch. Therefore, if we consider the extraction of effective sample pairs from a the whole dataset, it may indeed be attractive.

---

> ### Author Response · Authors · 2024-11-24
> **Further Response to Reviewer i7yC**
>
> We sincerely thank Reviewer i7yC for your thoughtful feedback on the benefits of RPO compared to existing methods, your alternative explanation from the runtime data augmentation perspective, and your suggestion for additional ablation studies.
>
> 1. First and foremost, we agree that runtime data augmentation could be a key factor in RPO’s effectiveness. It enables more efficient utilization of data within each GPU batch while incurring negligible computational and memory overhead. We also greatly appreciate your insightful comment: “Due to the high sparsity of text sequences, any method that increases data density in the text sequence space will be effective. This is also one of the important research directions for LLM at present.”
>
>    - While we fully agree with this perspective, our investigation suggests that increasing data density alone does not fully account for the effectiveness of RPO. This can be verified through experiments, as you suggested, which align with the ablation study presented in Table 1 of our manuscript. In this study, we analyzed Diagonal Weighting and Uniform Weighting under RPO. As shown in Table 1 of the manuscript, these methods perform reasonably well but fail to outperform DPO, let alone RPO with embedding-distance-based reweighting.
>
>
>     - Regarding the specific example you hypothesized, we believe you are referring to a scenario where data is augmented by consistently treating the winning response of one prompt as the losing response of a different prompt. While this setting has not been explored by any prior methods, RPO can indeed be applied directly using diagonal, uniform, or embedding-based reweighting.
>     - Although we have not conducted experiments specifically for this scenario, it aligns closely with the unpaired setting where we contrast the winning response of a prompt with losing responses from other prompts. In this context, our results clearly outperform KTO, a method specifically designed to address this unpaired setting.
>
>    - These findings indicate that while runtime data augmentation and increased data density in the text sequence space are valuable and worth emphasizing, they are not sufficient to surpass DPO in paired settings. The key lies in combining these concepts—an integral part of RPO’s innovation—with the proposed embedding-distance-based reweighting, which is critical to RPO’s success.
>
>     - We greatly appreciate your insights and hope this response effectively addresses your concerns.
>
> 2. Thank you for acknowledging the previous misunderstanding and clarifying your concern that RPO as a training algorithm may be underexplored.
>
>     - We would like to emphasize that the current RPO solution is limited by the memory constraints of GPUs, restricting us to a per-GPU batch size of 8. Alleviating this constraint through more advanced computing platforms, improved implementations of distributed computing, and additional code-level optimizations could further enhance RPO's performance. However, we argue that the potential for further improvement does not diminish the value of our current solution. On the contrary, it is an encouraging signal that further efforts should be dedicated to the RPO framework to develop future solutions.
>    - Second, retrieval-augmented techniques could potentially be used to identify contrastive pairs. While these techniques are interesting and promising, they are beyond the scope of the current paper.
>    - Third, even under extreme constraints, such as a per-GPU batch size of 2, we retain the flexibility to adjust $\tau$. It is worth noting that we fixed $\tau = 0.5$ when responding with "Examples from specific mini-batches." However, $\tau$ can be tuned to control the level of contrast between related and unrelated pairs, further enhancing the adaptability of RPO.
>
>     - We appreciate your thoughtful feedback and hope this addresses your concerns effectively.
>
> 3.  Thank you for suggesting that we organize our paper from the perspective of data augmentation rather than training optimization. We appreciate your insight and how it highlights ways to better explain and illustrate our contributions. In our next revision, we will make an effort to emphasize the runtime data augmentation perspective more prominently. However, we hope the reviewer considers this suggestion as a matter of personal preference regarding the paper’s style rather than a fundamental weakness. If RPO can inspire future work to explore the data augmentation perspective and develop improved solutions, we hope you will agree that it is a valuable contribution worthy of publication in a top-tier conference like ICLR.

---

> > ### Author Response · Authors · 2024-12-02
> > **Follow up with Reviewer i7yC**
> >
> > Dear Reviewer i7yC,
> >
> > We sincerely appreciate the time and effort you dedicated to reviewing our paper, and the invaluable suggestions you provided.
> >
> > Regarding your concern about the role of runtime data augmentation in the effectiveness of RPO, we believe our further response effectively addresses this point.
> >
> > **Quick Summary:**
> >
> > - Runtime data augmentation is indeed a key factor in RPO’s effectiveness, enabling efficient data utilization with minimal overhead. However, ablation studies (Table 1) show that data density alone does not fully explain RPO’s success; embedding-distance-based reweighting plays a critical role.
> > - Retrieval-augmented techniques offer promising potential for identifying contrastive pairs within the RPO framework. While these techniques are beyond the scope of this paper, they present an exciting avenue for future research building upon RPO.
> >
> > We deeply value your thoughtful feedback. As the extended discussion period ends tomorrow, we hope our responses address your concerns, and we’re happy to clarify further if needed.

---

### Official Review · Reviewer_hL8X · 2024-11-03

**Soundness:** 2
**Presentation:** 3
**Contribution:** 2
**Rating:** 5
**Confidence:** 4

**Summary:**

This paper introduces Relative Preference Optimization (RPO), an enhancement to the Direct Preference Optimization (DPO) method for aligning large language models (LLMs) with user preferences. RPO extends DPO by considering both identical and related prompts, and it uses a contrastive weighting mechanism to handle a broader range of preference data. Experiments demonstrate that RPO outperforms baseline methods on dialogue, summarization, and general evaluation benchmarks.

**Strengths:**

1. RPO addresses a limitation of DPO by incorporating both identical and related prompts, thereby expanding the model's ability to learn from a more diverse set of data.

2. The paper is well-written and clearly explains the methodology and experimental results, making it easy for readers to understand the contributions and their implications.

**Weaknesses:**

1. The approach of using sentence embeddings to assess semantic similarity may not be sensitive to token-level differences. This could be a significant drawback in domains like code and mathematics, where even minor differences in tokens of prompts can lead to entirely different or even contradictory answers.

2. The proposed RPO may be seen as a form of data augmentation, where different question answer pairs are combined during the training process to generate more data pairs to improve model performance. This augmentation approach is common in traditional data augmentation fields.

3. The experiments could be further strengthened. For instance, it would be interesting to see if using the model being trained to compute sentence similarity would yield better results. Additionally, the paper should explore whether this simple method provides any benefits for larger models, such as 13B or more capable ones, to fully validate its effectiveness across different scales.

**Questions:**

Please refer to the Weaknesses.

---

> ### Author Response · Authors · 2024-11-22
> **Response to Reviewer hL8X**
>
> We thank Reviewer hL8X for providing detailed feedback and valuable suggestions. We provide further clarifications below.
>
> **W1:** Thanks for pointing this out. RPO's flexibility lies in its ability to adjust the temperature parameter $\tau$, which allows us to control the proportion of additional relative prompt-response pairs included. This adaptability makes it possible to fine-tune $\tau$ for different task types and datasets. For example, in domains like mathematics and code, where even minor token-level differences in prompts can lead to entirely different or contradictory answers, we can set $\tau$ to a very small value. In such cases, RPO would effectively degenerate into DPO, restricting comparisons to responses originating from the same prompt.
>
> For the tasks covered in this paper, such as general dialogue and summarization tasks, we have thoroughly validated the effectiveness of RPO through extensive experiments across various base models. Nevertheless, for future applications in domains like code or mathematics, incorporating a token-level-aware embedding model could complement our current framework. However, our current experiments focus on general dialogue tasks, and the results strongly support the effectiveness of using sentence similarity to compute weights in this context.
>
> **W2:** Thank you for pointing this out. The motivation for RPO stems from human learning processes, where individuals transfer knowledge from similar questions or mistakes they have encountered. By building contrastive relationships across similar prompts, RPO is designed to emulate this aspect of human learning. While it can be interpreted as a form of data augmentation or data strengthening, this interpretation does not diminish its significance as an effective and impactful method.
>
> We would like to emphasize that RPO, compared to DPO, does not introduce any additional data. Both RPO and DPO iterate through the same training dataset—consisting of 160,000 data samples (or 320,000 prompt-response pairs)—exactly once. RPO instead enhances the learning process by leveraging additional relationships derived from the existing data, making it a straightforward yet powerful approach.
>
>
> **W3-1 (Prompt Encoding Model)**:  Thank you for your question! One of the key advantages of RPO is its use of a very lightweight model, MiniLM-L6-v2 (only 22M in size), to encode prompts, which introduces almost no additional computational overhead. In contrast, fine-tuning models typically have sizes of 7B or larger, and using these larger models to encode prompts could result in significant computational costs. However, exploring the use of the fine-tuning model for encoding prompts could be an interesting direction for future experiments.
>
>
> **W3-2 (Stronger Models)**: As shown in Table 3 of the manuscript, we provide a comparison of RPO with baseline methods on Llama-2-13B, demonstrating its effectiveness on larger models. Additionally, in Table 2 (refer to the general response section), we present results of fine-tuning Gemma-2-9B-it (a highly capable RLHF model) with RPO integrated into SimPO. These results further highlight the effectiveness of RPO.

---

> > ### Author Response · Authors · 2024-12-02
> > **Follow up with Reviewer hL8X**
> >
> > Dear Reviewer hL8X,
> >
> > We are deeply grateful for your insightful suggestions!
> >
> > As the rebuttal period will end soon, please kindly let us know if you have any remaining questions so that we can address them during the remaining discussion period.
> >
> > If our responses have addressed your concerns, please could you re-evaluate our work based on the updated information?

---

### Official Review · Reviewer_7wmZ · 2024-11-04

**Soundness:** 3
**Presentation:** 3
**Contribution:** 3
**Rating:** 6
**Confidence:** 4

**Summary:**

The Relative Preference Optimization (RPO) method presents a novel approach to aligning preferences that significantly improves data efficiency and effectiveness.

**Strengths:**

The proposed RPO method introduces an innovative approach to preference alignment, showcasing enhanced data efficiency and effectiveness. By integrating a contrastive weighting mechanism, RPO significantly broadens the learning capabilities of language models, allowing them to process a wider range of preference data. The experimental results across various tasks, such as dialogue, summarization, and general evaluation benchmarks, highlight RPO's superior adaptability and effectiveness in optimizing language model performance according to user preferences.

**Weaknesses:**

The paper states that "DPO does not fully reflect the complex nature of human learning, which often involves understanding contrasting responses to not only identical but also similar questions." However, there appears to be a lack of explicit experimentation or detailed discussion in the manuscript to substantiate this claim. It would be beneficial for the authors to provide empirical evidence or theoretical elaboration to support this assertion, which would strengthen the argument for RPO's advantages over existing methods like DPO.

**Questions:**

- How long does the embedding preprocessing step take?
- In my understanding, RPO utilizes a loss function similar to DPO's, augmented with a data reweighting mechanism. Could this mechanism be integrated with other approaches such as IPO, KTO, or SimPO? If so, what would the potential benefits or challenges be in combining these methodologies?

---

> ### Author Response · Authors · 2024-11-22
> **Response to Reviewer 7wmZ**
>
> We thank Reviewer 7wmZ for the positive feedback and valuable suggestions. Below, we provide further clarifications:
>
> **W1:**  We conduct thorough experiments to validate the effectiveness of RPO. The strong benchmark performance supports our claim that the newly introduced relative comparison is effective. Contrastive learning within win-lose pairs derived from related prompts aligns well with natural human reasoning, as demonstrated in Figure 1.
>
> **Q1:** Since the model used for encoding prompts, all-MiniLM-L6-v2, is only 22M in size, the encoding time for each batch of prompts is extremely short, averaging just 0.3 seconds. Additionally, as shown in Figure 4, the training time of RPO is nearly identical to that of baseline methods such as DPO/IPO, indicating that the additional computational cost introduced by prompt encoding is negligible throughout the training process.
>
>
> **Q2:** As shown in Table 2 (kindly refer to the general response section), we present the results of integrating RPO with SimPO. The outcomes on gemma-2-9b-it and Llama-3-8B-Instruct demonstrate that RPO_SimPO consistently outperforms SimPO, achieving superior results.

---

> > ### Comment · Reviewer_7wmZ · 2024-11-27
> > **Response to Authors**
> >
> > After carefully reviewing your response regarding the weaknesses, I still find that there is insufficient evidence to support the argument you have made. Therefore, my score remains unchanged.

---

> > > ### Author Response · Authors · 2024-11-27
> > >
> > > Dear Reviewer 7wmZ,
> > >
> > > Thank you for your response to our rebuttal. Could you kindly elaborate on which specific arguments you found lacking in evidence?
> > >
> > > - We believe we have provided compelling support for the advantages of RPO over DPO in paired data settings, as well as its benefits over KTO in unpaired scenarios.
> > >
> > > - If you could point out the areas of concern more specifically, we would be happy to clarify and explicitly highlight the relevant evidence.
> > >
> > > Thank you for your time and consideration.

---

### Official Review · Reviewer_2Go9 · 2024-11-09

**Soundness:** 3
**Presentation:** 3
**Contribution:** 3
**Rating:** 5
**Confidence:** 5

**Summary:**

The authors proposed new preference optimization methods to improve the performance of preference learning on various tasks. They explained that the current state-of-the-art methods cannot fully reflect the complex nature of human learning. To overcome this challenge, they proposed Relative Preference Optimization (RPO) to discern between more and less preferred responses derived from identical and related prompts. They showed that RPO outperforms DPO, KTO, and IPO on dialogue, summarization, and general evaluation benchmarks.

**Strengths:**

The authors explored one of the important problems of direct preference optimization methods, and it's interesting. Another advantage of this paper is the comprehensive experiment and theory used to satisfy their hypothesis.

**Weaknesses:**

I would appreciate that if the authors could clarify the following points:

- The results show that RPO outperforms other methods. I looked at the implantation they put in Appendix G and understood the main difference between RPO loss and DPO loss is the weight matrix. I am wondering how the authors can describe if a policy model has knowledge about the difference between the response of prompt i and j (assume prompt i is a reasoning question, and prompt j is a usual question that asks about the weather) will have a better performance?

- The Experiment section shows that the authors didn't explore the hyperparameters for other methods. To the best of my knowledge for the authors, direct methods like DPO, IPO, and KTO have a different performance on different hyper-parameters like Batch-Size, Beta, Regularization terms in IPO etc. Also, they didn't use beta=0.01 for DPO, which is the best hyper-parameter for this method. So, I suggest the authors compare their methods with the best DPO, KTO, and IPO hyperparameters.

- With respect to the comprehensive experiments, it seems the authors didn't report the scores for all models. For example, the experiments on Summarization are on the Mistral model, but for Anthropic-HH, it is on LLaMA (7B, 13B) and Mistral. Also, for other benchmarks, they didn't report the score for other methods. For example, on Open LLM  Leaderboard benchmarks, they just compared with KTO; another comparison method can be DRO [1]. I am confused as to why the authors used the Binarized Capybara Dataset because Ultrafeedback_binarized [2] is available, and they could use this dataset.

- New methods like SimPO and CPO are better choices for comparison. I encourage the authors to compare the proposed method with these algorithms, too.

- Using new models like LLaMA-3 can improve the experiment section.

---
[1] DRO:  https://arxiv.org/abs/2405.19107

[2] Ultrafeedback_binarized: https://huggingface.co/datasets/HuggingFaceH4/ultrafeedback_binarized

**Questions:**

All the points are mentioned in the Weakness section.

---

> ### Author Response · Authors · 2024-11-22
> **Response to Reviewer 2Go9**
>
> We thank Reviewer 2Go9 for providing detailed feedback and valuable suggestions. We provide further clarifications below.
>
> **W1**: Unlike DPO, which only considers win-lose contrasts within the same pair, RPO enables the construction of additional contrast lines for every prompt pair. As the reviewer pointed out, if two prompts $ i $ and $ j $ are significantly different, our reweighting technique ensures that the contrast line is heavily downweighted. This is because the answers to prompts $ i $ and $ j $ cannot be meaningfully extended to one another, as $ i $ and $ j $ are semantically unrelated.
>
> **W2**: Thank you for pointing this out. However, we would like to clarify that the optimal value of beta may vary depending on the base model and dataset. For the tasks covered in this paper—dialogue tasks on the HH dataset and summarization tasks on the summarization dataset—the baseline hyperparameters, such as beta, follow the default experimental settings provided by KTO. Please refer to the configuration at: [https://github.com/ContextualAI/HALOs/tree/main/config/loss](https://github.com/ContextualAI/HALOs/tree/main/config/loss). Specifically, the default value for beta is set to 0.1. Regarding the batch size, all experiments default to a per-GPU batch size of 8, which also aligns with KTO's default settings. Please see: [https://github.com/ContextualAI/HALOs/blob/main/config/model/base_model.yaml#L40](https://github.com/ContextualAI/HALOs/blob/main/config/model/base_model.yaml#L40).
>
> In our updated experiments (fine-tuning Meta-Llama-3-8B-Instruct on a general chat task with on-policy ultrafeedback dataset), we followed the settings of SimPO. In these experiments, all methods were tuned to their optimal hyperparameters, and our approach still achieved superior results. Please refer to Table 1(refer to the general response section) for details.
>
>
> **W3-1 (scores for other models)**: As mentioned in Line 471, the Mistral-7B model emerged as the top performer in our preliminary experiments, which led us to select it for detailed studies in the Summarization task. Since the OpenLLM leaderboard primarily evaluates models based on knowledge reasoning capabilities, and the HH dataset and Summarization dataset focus on assessing models' abilities to generate helpful and harmless responses and produce high-quality summaries, respectively, we did not evaluate our approach on the OpenLLM leaderboard.
>
> However, in our updated experiments (fine-tuning Meta-Llama-3-8B-Instruct on a general chat task with an on-policy ultra-feedback dataset), we conducted a detailed comparison with baselines, including results from the OpenLLM leaderboard. Please refer to Table 1 (kindly refer to the general response section) for further details.
>
> **W3-2 (Compare to DRO)**: Since DRO conducted experiments using the T5 model, their paper did not provide directly comparable results. Additionally, the method is not open-sourced, making it difficult for us to reproduce their results. Furthermore, DRO represents a framework that is independent of DPO/SimPO and similar approaches. As such, we believe it is appropriate to exclude DRO as a baseline for our work.
>
>
> **W3-3 (Comparison with KTO using unpaired Capybara and Ultrafeedback datasets)**: Thanks for pointing this out. Here, we aim to investigate the performance of our method in the **unpaired dataset** setting compared to KTO. The Ultrafeedback dataset mentioned remains a paired setting (each item is a quadruple: $ (x, y_1, y_2, \text{label}) $. Additionally, the argilla/distilabel-capybara-kto-15k-binarized dataset is a high-quality multi-turn dialogue dataset, making it particularly suitable for evaluating our method in multi-turn scenarios.
>
> To address the reviewer's concern, we also updated our experiments by fine-tuning meta-llama/Meta-Llama-3-8B-Instruct on the unpaired argilla/ultrafeedback-binarized-preferences-cleaned-kto dataset (where each item is a triplet: $ (x, y, 0/1) $. The results, presented in Table 3, demonstrate that RPO continues to outperform KTO in this unpaired setting.
>
> **Table 3**
> | **Benchmark**     | **KTO** | **RPO** |
> |-----------------|--------|--------|
> | Arc Challenge   | 65.27  | 66.98  |
> | TruthfulQA      | 61.40  | 60.78  |
> | Winogrande      | 78.22  | 78.30  |
> | GSM8k           | 68.92  | 70.81  |
> | Hellaswag       | 82.16  | 80.70  |
> | MMLU            | 64.68  | 65.60  |
> | Average     | 70.11 | **70.53** |
> |                 |        |        |
> | AlpacaEval (LC)      | 23.10  | **28.92**  |
> | MT-Bench        | 7.02   | **7.11**   |
> | Arena-Hard      | 14.30  | **18.40**  |
>
>
>
> **W4** and **W5**: Please see Table 1.

---

> > ### Comment · Reviewer_2Go9 · 2024-11-22
> >
> > Thanks so much for the comprehensive experiments. The experiments in ultrafeedback-armorm were very good because now I can compare your method with others fairly. There are some points:
> >
> > 1. The results for LLaMA-3-8B-Instruct reported by SimPO paper (See Table 12 on page 29) show that AlpacaEval (LC(%)) for SimPO is 53.7%, but in Table 1, you reported 42.19%. The discrepancy is not limited to SimPO; we can see it for IPO and DPO.
> >
> > 2. Your explanation about defining a semantics matrix $w$ and multiplying it by the loss function could not convince me. Current papers like the Slef-Consistency Preference Optimization (https://arxiv.org/pdf/2411.04109?) method showed that a weight for each pair can help the model increase performance iteratively. However, I couldn't find a valid reason for your method.
> >
> > 3. Finding the optimal value is one of the main steps of comparison. Instead of KTO, I refer you to the SimPO paper to see that they find the optimal value per method, and then they compare the best of others with SimPO. To the best of my knowledge, preference optimization methods are very sensitive to hyperparameters. So, exploring the hyperparameters will have a greater effect on your study.
> >
> > 4. The main point of using different models is to show that the method is independent of the model. Generally, if you want to evaluate your method on three models, you should report all the benchmarks' scores, at least in the Appendix. Otherwise, a reader can think you just got the best performance on model x, so you didn't report the performance of other models. This restricts your method per model and has a lack of generalizability.
> >
> > 5. DRO and DQO (https://arxiv.org/abs/2410.09302) are two main methods for unpair comparison. I encourage the authors to make a strong study by comparing their methods with these methods.
> >
> > 6. Assume a dataset D={$x,y_w,y_l$} where $x$, $y_w$, and $y_l$ represent the input, preferred, and less preferred responses, respectively. For 50% of the $x$, you can choose $y_w$ as 1 represents the samples with good answers. For the other 50%, you can choose $y_l$ as 0, representing the samples with bad answers. So you are not limited to using UltraFeedback_binarize for unpair comparison. The main problem I have with the argilla/distilabel-capybara-kto-15k-binarized dataset is the size of the data, and there is no statistic behind that. Also, KTO compared their method based on one preference with other methods, and for this reason, they defined two different regularization terms for preferred loss and less preferred loss. If you apply the unpair part of your method on ultrafeedback, then a reader can compare your methods with others fairly.
> >
> >
> > I thank the authors again for their comprehensive experiment during the rebuttal period, but for the above reasons, specifically 1 and 2, I will not increase my score, and I think this method lacks theory and comparison. Best of luck for you.

---

> > > ### Author Response · Authors · 2024-11-22
> > > **Further Response to Reviewer 2Go9**
> > >
> > > We thank Reviewer 2Go9 for engaging with our rebuttal and providing additional feedback. However, we believe there are some misunderstandings and best reviewing practices that require further clarification or consideration. Additionally, we are perplexed by the shift in the reviewer's opinion from the original positive assessment—“The authors explored one of the important problems of direct preference optimization methods, and it's interesting. Another advantage of this paper is the comprehensive experiment and theory used to satisfy their hypothesis”—to the conflicting statement: “I think this method lacks theory and comparison.” Understanding the reasoning behind this change would help us better address your concerns and provide a more focused response.
> > >
> > > Additionally, we respectfully but strongly request the reviewer to reconsider the relevance of papers published on arXiv after the ICLR submission deadline. In our view, if these papers are indeed highly relevant, it would be more appropriate for them to cite our work, RPO, which has been freely available on OpenReview, rather than the reverse. Without carefully studying the timeline and order of publication, it becomes challenging to fairly attribute credit to the efforts of hardworking authors in a meaningful way.
> > >
> > > It is important to note that credit for ideas should not be disproportionately influenced by the amount of attention they receive on social media, whether generated by the authors themselves, others, or by the citations such attention may garner. Respecting the timeline of publication is essential to ensure that ideas are not unfairly overshadowed, misappropriated, or overlooked. Academia should uphold standards that prioritize fairness and integrity over external noise, ensuring that contributions are evaluated based on merit and proper chronology.
> > >
> > > In summary, we are perplexed by the expectation to compare and discuss papers that were posted on arXiv clearly after our submission, as well as the apparent shift in the reviewer's sentiment. We believe this may be an oversight and respectfully request that you reconsider your position in light of our clarification regarding the timeline, along with the following points:
> > >
> > > > The results differences.
> > >
> > > Thank you for pointing this out. We would like to clarify that the choice of endpoints can significantly impact the results. In the SimPO paper, the endpoint used was gpt-4-1106-preview, whereas in our work, we used the gpt-4o endpoint. This difference accounts for the observed score discrepancy.
> > > To further illustrate this point, we provide the results of the original base model google/gemma-2-9b-it evaluated using the gpt-4o endpoint, which yielded a win rate of 42.19—substantially lower than the 51.1 reported on the SimPO GitHub page ([gemma-2-9b-it-alpaca](https://github.com/princeton-nlp/SimPO/tree/main?tab=readme-ov-file#gemma)).
> > > However, we would like to emphasize that all results presented in Table 1 and Table 2 were evaluated under the same settings for consistency. We hope this explanation addresses your concerns.
> > >
> > >
> > >
> > > > Explanation of semantic matrix.
> > >
> > > We are uncertain why the reweighting of loss terms using a contrastive matrix is confusing to the reviewer. From the widely accepted perspective of contrastive learning, this approach feels intuitive and natural. We measure the similarity between prompts, and if they are closely related, we assume their responses are also correlated. Related prompt-response pairs can be jointly learned through transferable knowledge. Thus, a higher similarity value corresponds to a greater weight $w$. Could you please clarify which specific aspect you find unclear or unsound? This would help us address your concerns more effectively.
> > >
> > > > Optimal hyperparameters.
> > >
> > > When we completed the initial version of RPO, SimPO did not even exist. So in the earlier setting, we followed the KTO configuration on the HH and Summarization datasets. Additionally, we conducted some ablations, as detailed in Appendix E.
> > > In the General Response section, Tables 1 and 2 present comprehensive experiments on the latest Llama3-8b-instruct model. In these experiments, all hyperparameters were fully aligned with the SimPO settings. Even with all baselines optimized to their best configurations, RPO consistently demonstrates superior performance.

---

> > > > ### Author Response · Authors · 2024-11-22
> > > > **Further Response to Reviewer 2Go9**
> > > >
> > > > > More Resuts.
> > > >
> > > > Thank you for your suggestion. In our earlier experiments, we followed DPO and KTO, conducting evaluations on LLaMA-2 and Mistral-7B to validate our method. As the field of preference alignment evolves, more models and datasets have emerged. We kindly direct the reviewer’s attention to the latest results in Tables 1 and 2, which were conducted on stronger base models (Llama-3-8b-instruct and Gemma-2-9b-it) and larger datasets (Ultrafeedback). These setups align with the current standards in preference alignment research. Notably, RPO continues to demonstrate excellent performance under these updated settings.
> > > > We will ensure that these latest results are included in the updated version of the paper.
> > > >
> > > >
> > > > > Compare with DRO and DQO.
> > > >
> > > > We would like to kindly remind the reviewer that DQO was posted on arXiv on October 11, 2024, which is after the ICLR submission deadline.
> > > > As for the DRO method, the experiments in the paper were conducted on the T5 model, making it difficult to directly compare results from the paper. Furthermore, the DRO work is not open-sourced and involves iterative training of both the value network and policy network. We welcome any suggestions from the reviewer on how best to conduct a fair comparison in this circumstance.
> > > >
> > > > > Ultrafeedback.
> > > >
> > > > We kindly ask the reviewer to revisit our response to W3-3. In this follow-up, we conducted additional experiments on the argilla/ultrafeedback-binarized-preferences-cleaned-kto dataset, which was derived from Ultrafeedback_binarized (paired) by converting it into an unpaired format. Our results demonstrate that RPO still achieves better performance than KTO on this dataset.

---

> > > > > ### Comment · Reviewer_2Go9 · 2024-11-23
> > > > >
> > > > > Thanks for the clarification. Unfortunately, I cannot find that where you explain using the GPT-4o as Judger. I assumed you used the default models for AlpacaEval, ArenaHard, and MT-Bench. Because the regular model for the AlpacaEval and ArenaHard benchmarks is GPT-4-turbo and for MT-Bench is GPT-4, I cannot evaluate the reported scores. Furthermore, your results contradict the findings of the SimPO paper. Your results show that SimPO is even worse than DPO. I am happy to see the explanation of the authors for this.
> > > > >
> > > > >
> > > > > I mentioned the DQO just to improve the comparison of the unpair method with the recent papers. DRO arXived 29 May 2024. I am looking for a valid explanation as to **why you didn't compare DRO with your method in terms of unpair method.**
> > > > >
> > > > > Also, Table 3 needs to be completed by adding the results of other models for AlpcaEval and Summarization tasks.
> > > > >
> > > > > ---
> > > > > Your response is not kind. However, I need to clarify that:
> > > > > "The authors explored one of the important problems of direct preference optimization methods, and it's interesting." My point is about proposing a method for one preference optimization. Still, I believe this part is interesting.
> > > > >
> > > > > "Another advantage of this paper is the comprehensive experiment and theory used to satisfy their hypothesis.". You evaluated your methods on different models and benchmarks. But your comparison is not completed. Please read the third point in weakness.

---

> > > > > > ### Author Response · Authors · 2024-11-23
> > > > > > **Further Response to Reviewer 2Go9**
> > > > > >
> > > > > > We thank the reviewer for engaging with us in seeking clarifications, and we sincerely apologize if our tone came across as unkind. As mentioned, misunderstandings can occur (from both sides), and we genuinely value the opportunity to clarify and address any concerns.
> > > > > >
> > > > > > > Clarification on Endpoint Usage
> > > > > >
> > > > > > - We chose to use the gpt-4o endpoint because gpt-4o represents a newer, more powerful model compared to gpt-4-1106-preview [1].  Moreover, the gpt-4o endpoint is more cost-effective, making it the most economical and practical model accessible to us.
> > > > > >
> > > > > > - Regarding why SimPO performed worse than DPO under GPT-4o, we suspect that SimPO may have overfitted to the metrics used with GPT-4-1106-preview.
> > > > > > - To provide further clarification, we reevaluated the reproduced results of SimPO fine-tuning on LLaMA-8B-Instruct using the GPT-4-1106-preview endpoint. The AlpacaEval LC win rate we achieved was 51.61, which is now comparable to the results reported in the SimPO paper. This finding further supports the validity of the results we reported under GPT-4o.
> > > > > >
> > > > > > We hope this clarification addresses the reviewer's concerns and provides greater context for our decisions.
> > > > > >
> > > > > > [1] [Chatbot Arena Leaderboard](https://huggingface.co/spaces/lmarena-ai/chatbot-arena-leaderboard)
> > > > > >
> > > > > >
> > > > > > > More results.
> > > > > >
> > > > > > Thank you for your feedback. We would like to emphasize that we have conducted comprehensive comparisons with the current state-of-the-art (with optimal parameters) on the paired On-policy UltraFeedback dataset. Moreover, on the unpaired dataset (capybara/ultrafeedback), we have thoroughly compared our method with the current open-source approach, KTO, and demonstrated that our RPO algorithm can further enhance the performance of SimPO, as validated on both LLaMA-3-8B and Gemma-2-9B-it models.
> > > > > >
> > > > > > These experimental results, we believe, provide robust evidence of the effectiveness and reliability of RPO.
> > > > > > If you believe that conducting experiments on HH and Summarization tasks using the latest LLaMA-3-8B model, in addition to the results already provided on UltraFeedback, would offer compelling reasons to address your concerns, we will make an effort to include these results in Table 3, time permitting.
> > > > > >
> > > > > > > Compared with DRO.
> > > > > >
> > > > > > As noted in our response, DRO is not open-sourced, and we were unable to reproduce their results due to the lack of publicly available implementation and detailed experimental settings for LLaMA-3-8B. We would appreciate it if the reviewer could clarify exactly what they would like us to do in this scenario.

---

### Author Response · Authors · 2024-11-22
**Global Response to Reviewers**

We deeply appreciate all reviewers for their valuable suggestions. In this section, we’ve updated additional tabular results of common interests.

**Table 1:** We fine-tuned Meta-Llama-3-8B-Instruct on a general chat task using the on-policy ultrafeedback dataset princeton-nlp/llama3-ultrafeedback-armorm. Each baseline method was trained with the optimal hyperparameters adopted in SimPO. RPO demonstrates superior performance across almost all benchmarks.

**Table 2:** Building on SimPO, we incorporated RPO's cross-prompt contrastive learning mechanism to fine-tune gemma-2-9b-it and Meta-Llama-3-8B-Instruct on the on-policy ultrafeedback dataset, respectively. Similarly, each baseline method was trained using the optimal hyperparameters from SimPO. The combined method, SimPO_RPO, achieves superior performance compared to SimPO alone.

**Table 1**:
| **Benchmark**    | **DPO** | **KTO** | **IPO** | **CPO** | **SimPO** | **RPO** |
|-------------------|---------|---------|---------|---------|-----------|---------|
| Arc Challenge     | 64.68   | 63.40   | 63.31   | 63.99   | 67.24     | 66.38   |
| TruthfulQA        | 55.82   | 55.62   | 60.77   | 56.28   | 64.71     | 61.17   |
| Winogrande        | 76.40   | 76.24   | 74.35   | 75.93   | 74.51     | 76.01   |
| GSM8k             | 76.95   | 75.36   | 75.21   | 75.74   | 70.74     | 74.83   |
| Hellaswag         | 79.70   | 79.56   | 76.45   | 79.06   | 78.15     | 79.88   |
| MMLU              | 65.91   | 65.95   | 65.52   | 65.72   | 65.09     | 65.95   |
| Average       | 69.91 | 69.36 | 69.27 | 69.45 | 70.07 | **70.70** |
|                   |         |         |         |         |           |         |
| AlpacaEval (LC)   | 41.59   | 41.46   | 43.90   | 41.37   | 42.19     | **44.20**   |
| MT-Bench          | **7.49**    | 7.43    | 7.33    | 7.47    | 7.21      | 7.45    |
| Arena-Hard        | 44.20   | 33.50   | 36.90   | 40.90   | 37.20     | **45.60**   |

| Method | lr    | beta  |
|--------|-------|-------|
| DPO    | 3e-7  | 0.01  |
| KTO    | 5e-7  | 0.01  |
| IPO    | 7e-7  | 0.5   |
| CPO    | 6e-7  | 0.05  |
| SimPO  | 1e-6  | 10    |
| RPO    | 7e-7  | 0.01  |



**Table 2**:

| **Benchmark**      | **Llama-3-8B-Instruct (SimPO)** | **Llama-3-8B-Instruct (SimPO_RPO)** | **gemma-2-9b-it (SimPO)** | **gemma-2-9b-it (SimPO_RPO)** |
|--------------------|---------------------------------|------------------------------------|---------------------------|------------------------------|
| Arc Challenge      | 67.24                          | 66.47                             | 67.83                     | 70.65                       |
| TruthfulQA         | 64.71                          | 64.35                             | 59.43                     | 60.49                       |
| Winogrande         | 74.51                          | 75.53                             | 75.14                     | 75.93                       |
| GSM8k              | 70.74                          | 70.51                             | 79.38                     | 82.11                       |
| Hellaswag          | 78.15                          | 77.85                             | 68.39                     | 70.38                       |
| MMLU               | 65.09                          | 65.27                             | 71.75                     | 71.97                       |
| Average       | 70.07                      | **70.16**                         | 70.32                 | **71.92**                   |
|                    |                                 |                                    |                           |                              |
| AlpacaEval (LC)    | 42.19                          | **44.72**                             | 50.05                     | **52.09**                       |
| MT-Bench           | 7.21                           | **7.31**                              | 8.02                      | **8.07**                        |
| Arena-Hard         | 37.20                          | **38.80**                             | 62.90                     | **63.50**                       |

---

### Meta-Review · Area_Chair_UhYV · 2024-12-09

**Metareview:**

This paper proposes Relative Preference Optimization (RPO) which is designed to discern between responses derived from identical and related prompts, which enables utilizing both paired and unpaired data. While the reviewers find the method to be interesting, they also raised significant concerns regarding the soundness of the method as well as the experiments/evaluations. The authors should consider reviewer's feedback when revising the manuscript.

**Additional Comments On Reviewer Discussion:**

The reviewers reached a consensus on rejection.

---

### Decision · Program_Chairs · 2025-01-22

Reject